# Beyond Majority Voting: LLM Aggregation by Leveraging Higher-Order Information

## Abstract

With the rapid progress of multi-agent large language model (LLM) reasoning, how to effectively aggregate answers from multiple LLMs has emerged as a fundamental challenge. Standard majority voting treats all answers equally, failing to consider latent heterogeneity and correlation across models. In this work, we design two new aggregation algorithms called Optimal Weight (OW) and Inverse Surprising Popularity (ISP), leveraging both first-order and second-order information. Our theoretical analysis shows these methods provably mitigate inherent limitations of majority voting under mild assumptions, leading to more reliable collective decisions. We empirically validate our algorithms on synthetic datasets, popular LLM fine-tuning benchmarks such as UltraFeedback and MMLU, and a real-world healthcare setting ARMMAN. Across all cases, our methods consistently outperform majority voting, offering both practical performance gains and conceptual insights for the design of robust multi-agent LLM pipelines.

## 1 Introduction

The aggregation of responses from multiple large language models has been widely used in practice. For example, a popular application is to improve reasoning via multi-agent LLM debate (Khan et al., 2024; Subramaniam et al., 2025; Choi et al., 2025) and LLM council (Zhao et al., 2024). Previous works thus far have mostly employed the simple *majority voting* (MV) rule as a natural first instinct to aggregate different LLMs' responses into a single answer. Intuitively, MV can be viewed as a zero-order aggregation method that only depends on the observed answers and fails to account for heterogeneity and correlation among models, which are often captured by higher-order information such as LLMs' expected accuracies (first-order information) and answer correlation (second-order information). This thus raises the following natural question: *is it possible to leverage such higher-order information to develop better methods for aggregating LLMs' responses?*

Effective solutions to the above problem not only can be applied to improve LLM debate, but also can help us understand the *collective* power frontier LLMs have in important tasks such as reasoning and forecasting. Such advances can be applied to many downstream applications of LLMs, such as making a profit in prediction markets (Karger et al., 2024) and enabling prediction as a service using LLMs (Santhosh et al., 2019; Peng et al., 2025). Notably, a relevant yet crucially different problem is the mixture of experts (MoE) method (Dai et al., 2024). MoE tackles a distinct problem where each expert is good at solving problems from certain domains, and their task is to learn which expert is the best to use upon seeing a query. Our response aggregation problem is different and requires fundamentally different techniques.[1]

Information aggregation is a natural and classic research question (Penrose, 1946; Tullock, 1959; Austen-Smith & Banks, 1996). At its core, it is the problem of aggregating random answers or predictions from different agents by accounting for their strengths while mitigating the random noise in their predictions. This general problem has been widely studied in both economic and machine learning literature (Austen-Smith & Banks, 1996; Frongillo et al., 2015; Guo et al., 2025), however, our focus on the LLM domain poses both new challenges and opportunities. On one hand, our problem of using LLMs for question answering gives rise to new problem structures such as symmetric

---

[1] In some sense, MoE aggregates "internal representations" via gated expert selection, rather than getting final outputs using higher-order information. Solutions to this scenario require intrinsically different techniques.

prior beliefs on correct and incorrect answers by assuming all problems' answers are randomly shuffled, as is typically done in practice. These problem structures allow us to derive an interpretable closed-form optimal solution to the problem tailored for LLM aggregation, which is otherwise impossible under general information structures. On the other hand, our focus on LLM applications also gives rise to new challenges, such as the high cost of obtaining the first-order information of expected accuracy, whose evaluation requires knowing many ground-truth answers. These challenges motivate us to study novel research questions, such as how to avoid the dependence on data's correct labels, and instead take advantage of the significantly lower cost of querying different LLMs on the same question so as to learn their correlations, i.e., the second-order information.

**Overview of Results.** Motivated by the wide use of majority voting for LLM aggregation in recent literature, this paper studies how to leverage higher-order information to better aggregate $N$ different LLMs' answers into a single answer. To obtain a principled understanding of this problem, we put forward a formal model rooted in the rich information aggregation literature (Austen-Smith & Banks, 1996; Frongillo et al., 2015) yet with novel ingredients directly motivated by LLM applications. We then proceed to study LLM aggregation schemes. First, when the designer knows all the LLMs' expected accuracy, we design a linear weighted aggregation scheme coined Optimal Weight (OW), which uses weight $w_i$ for LLM $i$ as the *inverse* function value of the sigmoid-like function $\sigma_K(x) = \frac{e^x}{K-1+e^x}$ on LLM $i$'s accuracy $x_i$. Perhaps surprisingly, this simple linear aggregator with carefully designed weights turns out to be the Bayesian-optimal aggregator — that is, it maximizes the expected accuracy among all possible (linear or not) aggregators with the same information access. We further show that this Bayesian-optimal aggregator has strictly higher accuracy than any single LLM under mild assumptions.

Despite the nice properties, a key limitation of applying this Bayesian-optimal aggregator is the use of accuracy $x_i$ whose estimation needs comparison with correct answers over many questions and is very costly to obtain. We thus turn to the use of second-order information about correlations of different LLMs' answers on a given question. Such information can be estimated from querying LLMs on many questions, without the need to know the true answers to these questions. For our principled analysis, we design an aggregator coined Inverse Surprising Popularity (ISP) — a counterfactual variant of the seminal surprising popularity rule of Prelec et al. (2017) — which only uses second-order information yet provably has an advantage over majority voting.

Finally, we empirically evaluate our approaches over both simulated settings which exactly match our technical assumptions and real-world LLM experiments on standard benchmarks, namely UltraFeedback and MMLU, as well as a real-world maternal health dataset ARMMAN (Mate et al., 2022). Due to a lack of access to correct answers in all these data sets, our first-order-based optimal aggregator is not directly applicable; hence, we evaluate the second-order-based method ISP as well as heuristics that estimate each LLM's accuracy from second-order information and apply it to the first-order-based aggregator. Across all the evaluation settings, we consistently observe that the heuristic combination of the (principled) first-order and second-order approach obtains the best empirical performance, though each single approach generally outperforms majority voting.

## 1.1 RELATED WORKS

We summarize below two lines of existing literature pertinent to our work.

**Multi-agent LLM reasoning.** Li (2024); Elumar et al. (2025); Subramaniam et al. (2025) simply use majority voting to aggregate LLM response. Du et al. (2023); Lu et al. (2024); Wang et al. (2024) assign distinct roles to different LLMs, and then improve performance by exchanging response information among them. Other lines of work instead focus on how to select among candidate outputs Jiang et al. (2023). Moreover, Chen et al. (2023a); Fu et al. (2025) demonstrate that aggregation of model outputs using confidence scores leads to significant improvements in accuracy, while Tekin et al. (2024) tries to maximize model diversity. For a comprehensive overview, interested readers can refer to Chen et al. (2025), which surveys reasoning approaches with multiple LLMs. However, our work considers a different unsupervised learning setting, where no true labels are available, e.g., using different LLM agents to automatically annotate datasets. We propose a multi-agent LLM reasoning framework that leverages higher-order information structures, and we theoretically show that both first- and second-order information can effectively improve over zero-order majority voting, yielding optimal aggregation for one-round reasoning.

**Information aggregation.** Information aggregation concerns how to combine noisy judgments in order to recover the truth. This line of work ranges from simple mechanisms like averaging and majority voting (Penrose, 1946; Tullock, 1959), to more recent algorithmic perspectives (Lin & Chen, 2023; Guo et al., 2025). Arieli et al. (2018) further develops a framework for identifying robust aggregators that perform well in the worst case without knowledge of the underlying information structure. A complementary line of research leverages *second-order* information—agents' beliefs about others' answers—to improve aggregation. This thread originates with Bayesian Truth Serum (Prelec, 2004) and the "surprisingly popular" rule (Prelec et al., 2017), and has since expanded to incorporate second- (and higher-) order signals in finite and potentially heterogeneous groups, both through theoretical analyses (Palley & Soll, 2019; Chen et al., 2021; Palley & Satopää, 2023; Kong, 2024; Pan et al., 2024) and empirical studies (Wang et al., 2021; Martinie et al., 2020; Wilkening et al., 2022). Our work extends this literature to the setting of heterogeneous LLM experts by designing aggregation rules that exploit both first- and second-order information, thereby broadening the scope of information aggregation research beyond human judgments.

Due to the space limit, we discuss these and more related works thoroughly in Appendix A.1.

## 2 MODEL

We consider a setting with a collection of questions without ground truth (e.g., an unlabeled dataset), and aim to predict, for each question, the correct answer from $K$ candidates (e.g., multiple-choice options; or for pairwise preference, $K = 2$). Assume $K$ is the same for all questions. Since we can assign a label to each candidate answer (e.g., $A, B, C, D$, etc.), we will focus on labels rather than the specific answer content throughout the paper. Therefore, let $S = \{s_1, \ldots, s_K\}$ denote the $K$ labels of possible answers. We can query from $N$ different LLM agents, i.e., different zero-shot LLMs in our setting, and observe their predicted answers across the entire collection of questions.

We build an information structure for the whole system. Assume we randomly draw one question uniformly, let $S^* \in S$ denote the ground-truth label of this problem. Also, let $A_1, \cdots, A_N \in S^N$ denote the predictions from $N$ agents. Note that $S^*, A_1, \cdots, A_N$ are all random variables. The correct answers to all questions are treated as fixed but unknown; the randomness here comes from drawing the problem and from the agents' prediction mechanisms. There is an underlying joint distribution $\mathbb{P}_0 \in \Delta(S \times S^N)$ of the ground-truth label and all agents' predictions. Agents may have heterogeneous accuracies, denoted by $x_1, \cdots, x_N$, such that $x_i = \mathbb{P}_0[A_i = S^*]$ for all $i \in [N]$. We consider the *prior-free* scenario that we have no knowledge about $\mathbb{P}_0$.

Our goal is to design aggregation algorithms $f : S^N \to \Delta(S)$ that better exploit cross-agent information to achieve higher accuracy than majority voting. Since we have no knowledge of the marginal distribution $\mathbb{P}_0(S^*)$, before introducing any specific aggregation algorithm, we first apply a pre-processing step to randomly shuffle all labels $s_1, \cdots, s_K$ for each question in the dataset. Besides being a standard practice, such random shuffling of labels also leads to more convenient proofs of guarantees for our aggregation algorithms and, under certain circumstances, is shown to be a transformation without information loss (see Appendix B.1). Let $\mathbb{P} \in \Delta(S^{N+1})$ denote the joint distribution after a random shuffle $\pi$.

We assume outputs of all LLM agents are not affected by the ordering of the options. With the improvement of LLMs' long-context abilities, we assume that they no longer forget or exhibit bias toward earlier options (Guo & Vosoughi, 2024). For example, if we ask the LLM agent which answer is better, namely $A$, $B$, or $C$, the agent replies $A$ with probability $80\%$, and both $B$ and $C$ with probability $10\%$. When the order changes to $B$, $C$, or $A$, the distribution of responses should remain $10\%$ for the first and second options and $80\%$ for the third option. This assumption also ensures that our pre-processing step does not change the information structure, up to relabeling.

From randomly shuffling, we obtain the following properties of the joint distribution $\mathbb{P}$:

**Proposition 1.** *The joint distribution $\mathbb{P}$ satisfies the following properties:*

- $\mathbb{P}(S^* = s_j) = \frac{1}{K}, \quad \forall j \in [K]$;
- $\mathbb{P}(A_i = s_j | S^* = s_j) = \mathbb{P}(A_i = S^*) = x_i, \quad \forall j \in [K], i \in [N]$;
- $\mathbb{P}(A_i = s_k | S^* = s_j) = \frac{1-x_i}{K-1}, \quad \forall j \in [K], i \in [N], k \neq j$.

We also make the following assumption of conditional independence among different LLM agents, which is a standard assumption in the information aggregation literature (Prelec et al., 2017; Arieli et al., 2018; Schoenebeck & Tao, 2021; Kong, 2024; Pan et al., 2024). However, this assumption may not hold perfectly in the LLM setting, especially when questions vary in difficulty. We break this canonical assumption and extend all our theoretical results to a more general setting in Appendix C, and will verify the effectiveness of our methods in experiments where perfect conditional independence might not hold.

**Assumption 1.** *[Conditional independence] Conditional on the ground truth label $S^*$, the agents' predictions $A_1, \cdots, A_N$ are independent. Mathematically, $\mathbb{P}(A_1, ..., A_N | S^*) = \prod_{i=1}^N \mathbb{P}(A_i | S^*)$.*

## 3 Leveraging First-order Information

We begin by aggregating *first-order information*: LLM agents' accuracies $x_1, \ldots, x_N$. In this section, we treat these inputs as given, assuming each $x_i$ is correct; practical estimation is deferred to Section 5. For comparison, *majority voting* (MV) uses only *zero-order information*—the raw answers—and therefore ignores heterogeneity in agent capability by assigning every agent equal weight. Formally, MV outputs the label with the largest unweighted vote count.

$$f_{MV}(a_1, \ldots, a_N) = \arg\max_s \sum_{i=1}^N w_i \mathbb{1}\{a_i = s\}, \quad \text{where } w_i = \tfrac{1}{N} \text{ for } \forall i \in [N]. \tag{1}$$

A natural improvement is to design weights based on first-order information $x_1, \ldots, x_N$. To this end, we propose a new weighted linear aggregation algorithm, the *Optimal Weight (OW) Algorithm 1*. Mathematically, OW sets the weight for agent $i$ as $\omega_i = \sigma_K^{-1}(x_i)$, where $\sigma_K(x) = \frac{e^x}{K-1+e^x}$. We prove the Bayesian optimality of OW among all aggregation algorithms, not limited to linear weighting. Here Bayesian optimality refers to choosing, given the joint distribution $\mathbb{P}$ over random variables $S^*, A_1, \ldots, A_N$ and observed LLM agents' answers $a_1, \ldots, a_N$, the prediction of the correct label $S^*$ that maximizes the posterior $\mathbb{P}(S^* = \cdot | A_1 = a_1, \ldots, A_N = a_N)$. In other words, Bayesian optimality is measured with respect to the expected accuracy under the true distribution $\mathbb{P}$.

In OW, we first apply a random shuffle mapping $\pi$ obtained from data pre-processing and finally map the results back. For example, in the binary case, we may ask either "Do you prefer $A$ or $B$?" or "Do you prefer $B$ or $A$?" with equal probability; the corresponding mapping $\pi$ is then determined directly by the question. For simplicity of presentation, we omit the normalization of weights.

---

**Algorithm 1** Optimal Weight (OW) Algorithm

1: **Input:** Accuracies $x_1, ..., x_N$.
2: Pre-processing: Randomly shuffle candidate labels, obtaining the shuffle mapping $\pi$.
3: Observe predictions $a_1, ..., a_N$.
4: Aggregate via $f_{OW}(a_1, \cdots, a_N) = \arg\max_{s \in S} \sum_{i=1}^N \sigma_K^{-1}(x_i) \mathbb{1}\{a_i = s\}$.
5: **Output:** Map back using $\pi^{-1}$ and return $\pi^{-1}(f_{OW}(a_1, \cdots, a_N))$.

---

**Theorem 1.** *Under Assumption 1, $f_{OW}$ defined in Algorithm 1 is the Bayesian optimal aggregator for any $\mathbb{P}$.*

For preference selection where there are only $K = 2$ labels, we have the following corollary.

**Corollary 1.** *When there are $K = 2$ labels, the optimal weights satisfy $\omega_i \propto \sigma^{-1}(x_i)$, where $\sigma(x) = \frac{e^x}{1+e^x}$ is the logistic function.*

Corollary 1 establishes a connection between the optimal weighting scheme and the ubiquitous Bradley–Terry (BT) model used in LLM post-training (Bradley & Terry, 1952; Ouyang et al., 2022; Chiang et al., 2024), thereby providing a theoretical justification for the validity of the BT model. This suggests that when combining agents with different capabilities, for instance, large models with higher accuracy but higher cost and small models with lower accuracy but lower cost, it is advisable to adopt an inverse-logistic weighting scheme, or more rigorously, Algorithm 1.

In addition, Theorem 1 characterizes the conditions under which majority voting is the optimal weighting strategy. When we repeatedly sample from the same LLM agent, for example, in com-

puting self-consistency within reasoning or fine-tuning (Wang et al., 2022; Chen et al., 2023b; Shao et al., 2024), majority voting is optimal.

**Corollary 2.** *When agents are homogeneous, i.e., $x_1 = x_2 = \cdots = x_N$, majority voting is the Bayesian optimal aggregator.*

It is also interesting to compare the Bayesian optimal aggregation algorithm OW with the best single agent. Excluding extreme cases, Algorithm 1 is strictly better than any individual agent used in aggregation. As shown in Proposition 2, this extreme case becomes harder to achieve as $K$ increases.

**Proposition 2.** *For any $\mathbb{P}$, let $f_i$ denote the aggregator that simply follows agent $i$'s prediction. Then $f_{OW}$ is strictly more accurate than $f_i$ if $\sigma_K^{-1}(x_i) < \sum_{j \neq i} \sigma_K^{-1}(x_j)$. In other words, unless there exists an exceptionally strong model such that $\sigma_K^{-1}(x_i) \geq \sum_{j \neq i} \sigma_K^{-1}(x_j)$, namely*

$$x_i \geq \frac{(K-1)^{N-2} \prod_{j \in [N] \setminus \{i\}} x_j}{(K-1)^{N-2} \prod_{j \in [N] \setminus \{i\}} x_j + \prod_{j \in [N] \setminus \{i\}} (1 - x_j)},$$

*Algorithm 1 strictly outperforms any individual agent.*

## 4 LEVERAGING SECOND-ORDER INFORMATION

One challenge of using first-order information is the need to estimate accuracies from the ground truth label $S^*$. However, knowing $S^*$ is often unrealistic. For example, in generalized unsupervised learning like automated data annotation (Li, 2024), the true label is never revealed.

In practice, the cost of generating outputs from LLMs is negligible compared to human annotation. Thus, we can easily obtain multiple samples $(A_1, \cdots, A_N)$, which motivates us to estimate and exploit information such as correlations between agents' predictions. For example, $\mathbb{P}(A_j | A_i)$ captures the distribution of agent $j$'s prediction as estimated by agent $i$ while forming its own prediction $A_i$. This type of information is referred to as *second-order information* (Prelec et al., 2017; Chen et al., 2021; Kong et al., 2022). We aim to leverage this second-order information in designing our aggregation algorithm. Note that the estimation of second-order information can be made arbitrarily accurate with a sufficiently large number of samples $(A_1, \cdots, A_N)$, without requiring any assumptions on the information structure $\mathbb{P}$. In what follows, we first present our results assuming an accurate $\mathbb{P}(A_j | A_i)$, and later discuss estimation with finite samples in Section 4.3.

Throughout, we assume that all agents perform no worse than random guessing, i.e., each agent achieves an accuracy of at least $\frac{1}{K}$. Otherwise, we can simply exclude inferior agents whose performance falls below that of a random predictor. After random shuffling, the second-order information exhibits desirable properties as a scoring rule, and we defer the discussion to Appendix E.1.

### 4.1 SURPRISINGLY POPULAR AGGREGATION

A seminal work of Prelec et al. (2017) introduced a method for aggregation based on second-order information, termed *surprising popularity (SP)*. For each agent, it estimates the distribution of other agents' predictions conditional on its own prediction. The option whose actual frequency exceeds the estimated frequency, i.e., the "surprisingly popular" option, is then selected.

We begin by following their framework. Mathematically, for every agent $i$ and candidate label $s$, we calculate the average probability that other agents believe agent $i$ will predict $s$, defined as the *score*

$$S_{SP}(s, i) = \frac{1}{N-1} \sum_{j \in [N] \setminus \{i\}} \mathbb{P}(A_i = s | A_j = a_j).$$

Then, SP selects the label that exhibits the largest positive gap relative to its predicted score:

$$f_{SP}(a_1, \ldots, a_N) = \arg\max_s \left( \sum_{i=1}^{N} \mathbb{1}\{a_i = s\} - \sum_{i=1}^{N} S_{SP}(s, i) \right). \tag{2}$$

We call the objective maximized by $f_{SP}$ the *advantage function*:

$$Adv_{SP}(s) = \sum_{i=1}^{N} \mathbb{1}\{a_i = s\} - \sum_{i=1}^{N} S_{SP}(s, i).$$

Note that the advantage function satisfies: $|Adv_{SP}(s)| \leq N$ and $\sum_{s \in S} Adv_{SP}(s) = 0$. Intuitively, a larger $Adv_{SP}(s^*)$ for the true label $s^*$ implies stronger distinguishing power of SP.

In Prelec et al. (2017), they show the power of SP when we have infinite homogeneous agents, that SP can always output the right label in that setting. But it is unclear how SP will perform in finite, heterogeneous settings. We then compare SP and the straightforward method, majority voting, finding that SP is actually not better than MV in this setting.

For $f_{MV}$ defined in Equation (1), we similarly define the advantage function as

$$Adv_{MV}(s) = \sum_{i=1}^{N} \mathbb{1}\{a_i = s\} - \frac{N}{K},$$

which also satisfies boundedness and $\sum_{s \in S} Adv_{MV}(s) = 0$, suggesting a fair comparison between $Adv_{MV}(s^*)$ and $Adv_{SP}(s^*)$. We further show that $\mathbb{E}[Adv_{MV}(s^*)] \geq \mathbb{E}[Adv_{SP}(s^*)]$, rigorously proved in Theorem 2. This implies MV is a more effective aggregation rule than SP in our setting.

This result is counterintuitive. In many human-subject settings, the majority answer is incorrect while the surprisingly popular answer coincides with the truth (Palley & Soll, 2019; Hosseini et al., 2021). This is because SP represents a fundamentally different aggregation perspective from MV. MV relies on distilling the "wisdom of the crowd". When most individuals vote for one answer, we tend to trust that answer as correct. In contrast, SP was proposed to correct systematic biases in common wisdom. It suggests that crowds tend to *underestimate* the probability of the ground-truth answer being chosen, and that by exploiting this bias, one can recover the correct answer.

Our setting of multi-agent LLMs, however, differs from that of human crowds. LLM agents are generally more powerful, so the systematic biases that SP exploits in human settings are much less pronounced here. As a result, the room for exploiting such bias is smaller, and the potential gain from doing so is dominated by the gain from aggregating common wisdom. This explains why SP does not outperform MV in this context.

## 4.2 FROM SURPRISINGLY POPULAR TO INVERSE SURPRISINGLY POPULAR

Motivated by the above explanation for why SP performs worse than MV, we propose a new scheme to improve SP, termed *Inverse Surprising Popularity (ISP)*, which builds on the intuition of amplifying prediction bias in a controlled way. We illustrate the intuition in the binary setting $S = \{s_1, s_2\}$ for the simplicity of presentation.

For SP, consider any agent $i$. Recall that the predicted score for $s^*$ is

$$S_{SP}(s^*, i) = \frac{1}{N-1} \sum_{j \in [N] \setminus \{i\}} \mathbb{P}(A_i = s^* | A_j = a_j).$$

Without loss of generality, consider another agent $j \neq i$ and assume $S^* = s_1$. Omitting $\frac{1}{N-1}$, its expected contribution to the score is

$$\mathbb{P}(A_i = s_1 | A_j = s_1)\,\mathbb{P}(A_j = s_1 | S^* = s_1) + \mathbb{P}(A_i = s_1 | A_j = s_2)\,\mathbb{P}(A_j = s_2 | S^* = s_1). \quad (3)$$

Ideally, a smaller score increases the advantage function of $s^*$. Note that

$$\mathbb{P}(A_i = s_2 | A_j = s_1) = \mathbb{P}(A_i = s_2 | A_j = s_1) \leq \mathbb{P}(A_i = s_2 | A_j = s_2) = \mathbb{P}(A_i = s_1 | A_j = s_1),$$

suggesting that humans tend to assign higher predictions to answers that match their own, compared to those that differ. Based on this, a natural alternative to Equation (3) is to consider

$$\mathbb{P}(A_i = s_1 | A_j = s_2)\,\mathbb{P}(A_j = s_1 | S^* = s_1) + \mathbb{P}(A_i = s_1 | A_j = s_1)\,\mathbb{P}(A_j = s_2 | S^* = s_1). \quad (4)$$

Therefore, we can summarize ISP as considering the prediction that would be made when agents report the other answers, which is intuitively more biased than the current prediction. Now we formally define ISP based on Equation (4). Similarly, we first define the ISP score as

$$S_{ISP}(s, i) = \frac{1}{N-1} \sum_{j \in [N] \setminus \{i\}} \frac{1}{K-1} \sum_{a \in S \setminus \{a_j\}} \mathbb{P}(A_i = s | A_j = a), \quad (5)$$

which yields the ISP aggregator and ISP's advantage as

$$f_{ISP}(a_1, \cdots, a_n) = \arg\max_s Adv_{ISP}(s) = \arg\max_s \sum_{i=1}^N \mathbb{1}\{a_i = s\} - \sum_{i=1}^N S_{ISP}(s, i). \quad (6)$$

The mechanism of the ISP aggregator is summarized in Algorithm 2.

---

**Algorithm 2** Inverse Surprising Popularity (ISP) Algorithm

---

1: **Input:** Second-order information after shuffling $\mathbb{P}(A_i = s_k | A_j = s_l)$ for every $(i, j, k, l)$.
2: Pre-processing: Randomly shuffle candidate labels, obtaining the shuffle mapping $\pi$.
3: Observe predictions $a_1, ..., a_N$.
4: Compute advantage $Adv_{ISP}$ using Equations (5) and (6).
5: Aggregate vis $f_{ISP}(a_1, \cdots, a_N) = \arg\max_{s \in S} Adv_{ISP}(s)$.
6: **Output:** Map back using $\pi^{-1}$ and return $\pi^{-1}(f_{ISP}(a_1, \cdots, a_N))$.

---

Example 1 presents a case of $K = 2$ where ISP outperforms MV, which in turn outperforms SP. We will later prove theoretically that this ordering of performance holds for general cases in expectation.

**Example 1.** *We assume there are 4 agents with accuracies $x_1 = x_2 = 1$ and $x_3 = x_4 = 0.5$. For MV, we break ties randomly. We assume the true label after shuffling is $s_1$ without loss of generality owing to symmetry. The results are in Table 1. We notice that MV aggregates the correct prediction with probability $\frac{7}{8}$, while SP is correct with probability $\frac{3}{4}$. ISP, however, always obtains the correct prediction. We give another Example 3 in Appendix E.2 for arbitrary ways of tie breaking.*

| $(A_1, A_2, A_3, A_4)$ | Probability | MV | SP | ISP |
|---|---|---|---|---|
| $(s_1, s_1, s_1, s_1)$ | 1/4 | $s_1$ | $s_1$ | $s_1$ |
| $(s_1, s_1, s_1, s_2)$ | 1/4 | $s_1$ | $s_1$ | $s_1$ |
| $(s_1, s_1, s_2, s_1)$ | 1/4 | $s_1$ | $s_1$ | $s_1$ |
| $(s_1, s_1, s_2, s_2)$ | 1/4 | $0.5 \circ s_1 + 0.5 \circ s_2$ | $s_2$ | $s_1$ |

Table 1: Aggregation results of MV, SP and ISP.

Note that MV, SP and ISP all select the candidate with maximum advantage, that is, they respectively maximize $Adv_{MV}$, $Adv_{SP}$ and $Adv_{ISP}$. Since for all three, the sum of advantages over all labels is zero by probability normalization, effective aggregation requires the correct label $s^*$ to attain the largest advantage. This observation provides the foundation for our subsequent theorem.

**Theorem 2.** *Under Assumption 1, ISP in Algorithm 2 outperforms MV, which in turn outperforms SP, in expectation. That is, $\mathbb{E}[Adv_{ISP}(s^*)] \geq \mathbb{E}[Adv_{MV}(s^*)] \geq \mathbb{E}[Adv_{SP}(s^*)]$. More specifically, it holds that*

$$\mathbb{E}[Adv_{ISP}(s^*) - Adv_{MV}(s^*)] = \frac{\sum_{i=1}^N \sum_{j \in [N] \setminus \{i\}} (Kx_i - 1)(Kx_j - 1)^2}{(N-1)K(K-1)^3},$$

*and*

$$\mathbb{E}[Adv_{MV}(s^*) - Adv_{SP}(s^*)] = \frac{\sum_{i=1}^N \sum_{j \in [N] \setminus \{i\}} (Kx_i - 1)(Kx_j - 1)^2}{(N-1)K(K-1)^2}.$$

In Theorem 2, as the number of options increases, ISP's superiority over MV gradually diminishes, as $\mathbb{E}[Adv_{ISP}(s^*) - Adv_{MV}(s^*)] \simeq \Theta(1/K)$. However, MV remains consistently better than SP, say $\mathbb{E}[Adv_{MV}(s^*) - Adv_{SP}(s^*)] \simeq \Theta(1)$ with respect to $K$. This effect arises from the exploration component in ISP, whose efficiency decreases as $K$ grows. In other words, when agents make incorrect predictions, exploring the correct label becomes harder. Hence, in practical settings where there are only a few options, ISP provides stronger guarantees and is more desirable to use.

## 4.3 PRACTICAL ESTIMATION OF SECOND-ORDER INFORMATION

We now return to the practical setting, where only finitely many samples are available, say a dataset consisting of $M$ questions and corresponding responses from $N$ LLM agents, to estimate the second-

order information $\mathbb{P}(A_i|A_j)$. We can naturally employ empirical conditional probabilities to estimate, that is,

$$\widehat{\mathbb{P}}(A_i = s_k | A_j = s_l) = \frac{\#\{A_i = s_k, A_j = s_l\}}{\#\{A_j = s_l\}}.$$

Although $\widehat{\mathbb{P}}$ and $A_1, ..., A_N$ are not independent, we still have the following theorem after random shuffling, showing that when data size $M$ is sufficiently large, the empirical advantage of ISP remains greater than that of MV.

**Theorem 3.** *Under Assumption 1, when there are $M$ i.i.d. questions, it holds that for every question, with probability at least $1 - \delta$,*

$$\mathbb{E}[\widehat{Adv}_{ISP}(s^*) - Adv_{MV}(s^*)] \gtrsim \frac{\sum_{i=1}^N \sum_{j \in [N] \setminus \{i\}} (Kx_i - 1)(Kx_j - 1)^2}{(N-1)K(K-1)^3} - \widetilde{\mathcal{O}}(\sqrt{\frac{1}{M} \log(\frac{1}{\delta})}),$$

*where $\widetilde{\mathcal{O}}(\cdot)$ suppresses some logarithmic factors in $M$. Here, the expectation is only taken over predictions $A_1, ..., A_N$ to this question and*

$$\widehat{Adv}_{ISP}(s^*) = \sum_{i=1}^N \mathbb{1}\{a_i = s^*\} - \sum_{i=1}^N \frac{1}{N-1} \sum_{j \in [N] \setminus \{i\}} \frac{1}{K-1} \sum_{a \in S \setminus \{a_j\}} \widehat{\mathbb{P}}(A_i = s^* | A_j = a).$$

## 5 EXPERIMENTS

We organize our experiments as follows. In Section 5.1, we use simulations to demonstrate the advantages of ISP over MV and SP. Section 5.2 introduces our methodology, showing how second-order information inspires the estimation of optimal weights in OW for unavailable accuracies. In Sections 5.3 and 5.4, we apply our methods to reinforcement learning with human feedback (RLHF) and healthcare datasets, illustrating how both OW and ISP can improve upon MV in practice.

### 5.1 EVALUATION ON SIMULATED DATA

We simulate a setting with $N = 4$ agents whose accuracies are fixed at 0.6, 0.7, 0.8, and 0.9, respectively, to mimic LLM performance heterogeneity. We randomly generate $M = 10,000$ questions and vary the number of answer choices $K \in \{2, 4, 6, 8, 10\}$. For MV, ties are broken uniformly at random. For SP and ISP, we leverage empirical second-order information to compute the advantage; for example, in Algorithm 2, we input $\widehat{\mathbb{P}}(A_i = s_k | A_j = s_l) = \frac{\#\{A_i=s_k, A_j=s_l\}}{\#\{A_j=s_l\}}$ within the dataset. We present our results in Table 2. We use OPT to denote the Bayesian optimal algorithm OW with clairvoyant access to the best weights, and Single Best to represent the strongest agent.

Table 2: Experimental Results: We report the overall accuracy for different values of $K$. In the tables, the best results are highlighted in **bold**, and the second-best results are underlined.

| Accuracy | $K = 2$ | $K = 4$ | $K = 6$ | $K = 8$ | $K = 10$ |
|---|---|---|---|---|---|
| MV | 85.13% | 92.64% | 94.22% | 94.85% | 95.54% |
| SP | 79.94% | 90.52% | 92.68% | 93.66% | 94.40% |
| Single Best | 90.34% | 89.94% | 90.31% | 89.95% | 90.05% |
| ISP (ours) | **90.48%** | **94.45%** | **95.78%** | **96.23%** | **96.49%** |
| OPT | 91.37% | 94.94% | 96.05% | 96.46% | 96.81% |

We clearly observe that ISP outperforms MV, which in turn outperforms SP, across distinct values of $K$, and the gap between ISP and MV vanishes as $K$ increases (cf. Figure 1 in Appendix F.1), empirically validating Theorem 2. Moreover, we find that ISP consistently outperforms the single best agent, showing that our LLM aggregation approach improves prediction accuracy beyond any individual model by exploiting higher-order information among agents.

### 5.2 ADAPTED OPTIMAL WEIGHT ALGORITHM WITHOUT GROUND TRUTH

The challenge of OW is that, in practice, we lack access to the accuracies of LLM agents. Second-order information not only enables the design of an alternative algorithm, ISP, but also inspires us

to estimate accuracies in an unsupervised manner. We employ two approaches to leverage second-order information as input to the OW algorithm, thereby improving LLM aggregation accuracy.

1) We note that the true second-order information is a function of the accuracies $x_1, ..., x_N$. By exploiting empirical second-order information, we can apply empirical risk minimization to learn the true accuracies. After estimating the accuracies, we can implement Algorithm 1 using the estimated optimal weights $\sigma_K^{-1}(\widehat{x}_i)$. We denote this OW with learning pipeline by *OW-L*. Mathematically,

$$\widehat{x}_1, ..., \widehat{x}_N = \underset{x_1,...,x_N}{\arg\min} \sum_{i,j,k,l} \Big( \mathbb{P}(A_i = s_k | A_j = s_l)[x_1, ..., x_N] - \widehat{\mathbb{P}}(A_i = s_k | A_j = s_l) \Big)^2. \quad (7)$$

The expanded expressions are presented in Appendix F.2.

2) Another approach is to use ISP to obtain aggregated predictions for each question. We then treat these predictions as pseudo ground truth to estimate the accuracies, and subsequently construct the optimal weights based on them. We denote this method as *OW-I* to distinguish it. Mathematically,

$$\widehat{x}_i = \frac{\#\{f_{ISP}(a_1, ..., a_N) = a_i\}}{M} \text{ for every } i.$$

In the next section, we implement these two approaches, OW-L and OW-I, on real datasets, without access to true accuracies of LLM agents. This shows that our aggregation algorithms improve upon simple majority voting, offering an information-theoretic view on multi-agent LLM reasoning.

### 5.3 REAL-WORLD DATASETS

**Models.** We employ four distinct families of large language models, and within each family select one relatively strong and one relatively weak model. Specifically, we use GPT = {GPT-4o-2024-11-20, GPT-35-turbo-0125} (Hurst et al., 2024), Qwen = {Qwen2.5-Instruct-14B, Qwen2.5-Instruct-3B} (Qwen et al., 2025), Llama = {Llama3.1-8B-Instruct, Llama3.2-1B-Instruct} (Dubey et al., 2024), and Phi = {Phi-4, Phi-4-mini-instruct} (Abdin et al., 2024), yielding in total 16 combinations.

**Datasets.** We first utilize two widely adopted fine-tuning datasets for large language models, UltraFeedback (Cui et al., 2023) and MMLU (Hendrycks et al., 2020). For UltraFeedback, we consider the case of $K = 2$. We leverage LLMs to predict preferences, and our results demonstrate that LLM agents can automatically provide labels for preference datasets with high accuracy. This substantially reduces the reliance on costly human annotation and offers a new perspective for scaling RLHF with LLMs. Each question in MMLU contains $K = 4$ choices. The LLM agents are required to select the best answer, which mimics real-world scenarios where LLMs must make decisions within a finite action space. This setting highlights that our OW and ISP algorithms can substantially improve upon majority voting when handling queries with a finite set of options.

We further apply our LLM aggregation framework in collaboration with the non-profit organization ARMMAN (Mate et al., 2022). Motivated by recent work on the promise of LLMs in predicting human behavior (Martinson et al., 2025), the task is to predict which pregnant and postpartum women are at risk of dropping out from call-based maternal health information programs, corresponding to the case of $K = 2$. The goal is to improve program retention and, ultimately, maternal health outcomes. This illustrates the potential of our algorithms to facilitate multi-agent LLM reasoning in critical healthcare settings. Additional details (e.g., computational resources and runtime) and prompts are provided in Appendix F.3.

### 5.4 EVALUATION RESULTS ON REAL-WORLD DATASETS

In this section, we present experimental results on UltraFeedback, MMLU and ARMMAN to investigate the roles of first-order and second-order information in our framework. Note that for real-world datasets, the relative ranking of model families is unknown; therefore, we only report the accuracy of the single best-performing model as a reference. Namely, Single Best functions as a clairvoyant oracle rather than a fair baseline for a comprehensive comparison. We compare our proposed methods, OW-L, OW-I, and ISP, against the widely used MV baseline. We select one model in each model family and obtain 16 possible model ensembles. In Table 3, we report aggregated

Table 3: Experimental Results: We report the overall accuracy on three datasets. In the tables, the best results are highlighted in **bold**.

| Accuracy | OW-L (ours) | OW-I (ours) | ISP (ours) | MV | Single Best |
|---|---|---|---|---|---|
| UltraFeedback | **73.66%** | **73.66%** | 73.26% | 72.21% | 73.14% |
| MMLU | **90.37%** | **90.37%** | 90.01% | 89.32% | 91.02% |
| ARMMAN | **85.78%** | **85.78%** | **85.78%** | 85.24% | 85.32% |

results for the strongest model within each family, namely GPT-4o-2024-11-20, Qwen2.5-Instruct-14B, Llama3.1-8B-Instruct and Phi-4, leaving the remaining combinations to Appendix F.4.

When using the four strong models, all three of our algorithms outperform majority voting. On the UltraFeedback, MMLU and ARMMAN datasets, the four models provided different answers for 52.09%, 31.29% and 46.52% of the questions, respectively. Any aggregation algorithm can only potentially improve accuracy on these questions, and on these subsets, our method respectively yields absolute accuracy gains of 2.78%, 3.36% and 1.16% over MV. Moreover, when selecting the strongest model from each family, our aggregation methods outperform all participating models on both UltraFeedback and ARMMAN. This result indicates LLM aggregation can extend the boundary of model capabilities and highlights the substantial potential of multi-agent LLM reasoning.

We further provide a per-question comparison in Table 4 and conduct hypothesis testing for three datasets to evaluate the performance of our methods (e.g., OW-I) and MV. The resulting t-statistics are 12.53 (UltraFeedback), 23.39 (MMLU), and 3.22 (ARMMAN), indicating that our aggregation methods are *significantly* better than MV.

| Discrepancy Counts | UltraFeedback | MMLU | ARMMAN |
|---|---|---|---|
| OW-L | 2545 / 1727 | 1821 / 659 | 264 / 195 |
| OW-I | 2545 / 1727 | 1821 / 659 | 264 / 195 |
| ISP | 2369 / 1762 | 1522 / 667 | 264 / 195 |

Table 4: Per-question comparison: MV wrong $\rightarrow$ Ours correct / Ours wrong $\rightarrow$ MV correct.

Across all 16 ensembles, OW-L outperforms MV in 97.92% of the cases, while OW-I and ISP outperform MV in 85.83% of the cases. MV never achieves the best performance in any case, and the absolute improvement in accuracy ranges from 0.54% to 14.20%. The best-performing algorithm sometimes varies by case. Accounting for ties where multiple methods achieve the same accuracy, OW-L attains the highest accuracy in 66.67% of the cases, OW-I in 72.92%, and ISP in 12.50%. This demonstrates leveraging higher-order information enables effective aggregation of LLM responses without any labels, providing a practical solution for multi-LLM reasoning in real-world settings.

## 6 CONCLUSION AND DISCUSSION

In this paper, we study how higher-order information can improve the performance of multi-agent LLM reasoning. Using first-order information, we propose a Bayesian-optimal aggregation scheme. For the unsupervised setting without any ground-truth labels, we show that leveraging second-order information across models can provably improve upon the widely used majority voting. Through simulations, LLM post-training datasets, and real-world healthcare data, we demonstrate that our OW and ISP algorithms consistently dominate majority voting. Our results provide new insights into the role of information in multi-agent LLM reasoning, with implications for applications such as LLM debates (Subramaniam et al., 2025).

Questions naturally arise for future exploration. How can contextual learning be leveraged to assign more fine-grained, prompt-specific weights based on expected accuracy? Can incorporating even higher-order information provide additional insights to further improve aggregation performance? To what extent might integrating our aggregation framework with LLM post-training push the boundaries of LLM capabilities (Cheng et al., 2024)? We leave these interesting questions as potential next steps.

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

# A OMITTED DETAILS IN SECTION 1

## A.1 RELATED WORKS

**Multi-agent LLM reasoning.** Unlike LLM mixture-of-experts (MoE) (Jacobs et al., 1991; Jiang et al., 2024; Team, 2024; Liu et al., 2024; Dai et al., 2024), which is primarily motivated by engineering considerations, LLM aggregation focuses on ensembling the answers of different LLM agents in order to improve accuracy. Li (2024); Elumar et al. (2025); Subramaniam et al. (2025) simply use majority voting to aggregate. Du et al. (2023); Lu et al. (2024); Wang et al. (2024) assign distinct roles to different LLMs, and then improve performance by exchanging response information among them. Other lines of work instead focus on how to select among candidate outputs Jiang et al. (2023). Moreover, Chen et al. (2023a); Fu et al. (2025) demonstrate that aggregation of model outputs using confidence scores leads to significant improvements in accuracy, while Tekin et al. (2024) tries to maximize model diversity. Fang et al. (2024) applies LLM aggregation to the e-commerce setting and learns optimal LLM weights based on product attributes. For a comprehensive overview, interested readers can refer to Chen et al. (2025), which surveys reasoning approaches with multiple LLMs. However, our work considers a different unsupervised learning setting, where no true labels are available, e.g., using different LLM agents to automatically annotate datasets. We propose a multi-agent LLM reasoning framework that leverages higher-order information structures, and we theoretically show that both first- and second-order information can effectively improve over zero-order majority voting, yielding optimal aggregation for one-round reasoning.

**Information aggregation.** Information aggregation concerns how to combine noisy judgments in order to recover the truth. This line of work ranges from simple mechanisms like averaging and majority voting (Penrose, 1946; Tullock, 1959), to more recent algorithmic perspectives (Lin & Chen, 2023; Guo et al., 2025). Arieli et al. (2018) further develops a framework for identifying robust aggregators that perform well in the worst case without knowledge of the underlying information structure. Neyman & Roughgarden (2022) extends this setting to continuous domains and shows that, under certain families, simple averaging is sufficiently good. De Oliveira et al. (2021); Levy & Razin (2022); Arieli et al. (2025) focus on environments with unknown correlations but known marginal distributions. A complementary line of research leverages *second-order* information—agents' beliefs about others' answers—to improve aggregation. This thread originates with Bayesian Truth Serum (Prelec, 2004) and the "surprisingly popular" rule (Prelec et al., 2017), and has since expanded to incorporate second- (and higher-) order signals in finite and potentially heterogeneous groups, both through theoretical analyses (Palley & Soll, 2019; Chen et al., 2021; Palley & Satopää, 2023; Kong, 2024; Pan et al., 2024) and empirical studies (Wang et al., 2021; Martinie et al., 2020; Wilkening et al., 2022). Our work extends this literature to the setting of heterogeneous LLM experts by designing aggregation rules that exploit both first- and second-order information, thereby broadening the scope of information aggregation research beyond human judgments.

# B OMITTED DETAILS IN SECTION 2

## B.1 JUSTIFICATION OF RANDOM SHUFFLING

For the simplicity of presentation, we use $K = 2$ with $S = (s_1, s_2)$ to show the result, and then discuss how to extend it easily to a general $K$.

Define two function families: $F_{large}$ is the original function family of all possible aggregation rules $f : S^N \to \Delta(S)$; $F_{small} \subseteq F_{large}$ denotes all functions $f$ satisfying

$$f(a_1, \cdots, a_N) + f(1 - a_1, \cdots, 1 - a_N) = 1, \quad \text{for all } a_1, \cdots, a_N.$$

Note that $f$ can be randomized, so we slightly abuse notation and use $f(a_1, \cdots, a_N)$ to denote the probability that $f$ outputs $s_1$.

We use $P_{large}$ to denote a large joint distribution family that satisfies:

- Symmetry: For all $i$, $\mathbb{P}(A_i = s_1 \mid S^* = s_1) = \mathbb{P}(A_i = s_2 \mid S^* = s_2)$.
- Conditional Independence: $\mathbb{P}(A_1, \cdots, A_N \mid S^*) = \prod_{i=1}^{N} \mathbb{P}(A_i \mid S^*)$.

We use $P_{small} \subseteq P_{large}$ to denote those $\mathbb{P}$ that also satisfy the balance condition $\mathbb{P}(S^* = s_1) = \mathbb{P}(S^* = s_2) = 1/2$.

Assume we only know that the information structure $\mathbb{P} \in P_{large}$ and we hope to find a good aggregator. We hope to consider the worst-case scenario since we have no extra knowledge. The performance is

$$\min_{f \in F_{large}} \max_{\mathbb{P} \in P_{large}} L(f, \mathbb{P}),$$

where $L(f, \mathbb{P})$ is the error probability of $f$ under $\mathbb{P}$:

$$L(f, \mathbb{P}) = \mathbb{E}_{(S^*, A_1, \cdots, A_N) \sim \mathbb{P}} \mathbb{1}\{f(A_1, \cdots, A_N) \neq S^*\}.$$

If we perform random shuffle, the performance is analyzed as

$$\min_{f \in F_{large}} \max_{\mathbb{P} \in P_{small}} L(f, \mathbb{P}).$$

For any aggregation function $f(a_1, \cdots, a_N)$, define the symmetric function

$$\bar{f}(a_1, \cdots, a_N) = 1 - f(1 - a_1, \cdots, 1 - a_N).$$

Equivalently, random shuffling can be seen as using $f$ with probability $1/2$ and $\bar{f}$ with probability $1/2$. The resulting symmetrized rule lies in $F_{small}$, so we can also write

$$\min_{f \in F_{small}} \max_{\mathbb{P} \in P_{large}} L(f, \mathbb{P}).$$

Since $F_{small} \subseteq F_{large}$ and $P_{small} \subseteq P_{large}$, we always have

$$\min_{f \in F_{small}} \max_{\mathbb{P} \in P_{large}} L(f, \mathbb{P}) \geq \min_{f \in F_{large}} \max_{\mathbb{P} \in P_{large}} L(f, \mathbb{P}) \geq \min_{f \in F_{large}} \max_{\mathbb{P} \in P_{small}} L(f, \mathbb{P}).$$

On the other hand, viewing random shuffling as mixing $f$ and $\bar{f}$ shows that optimizing over $F_{small}$ against $P_{large}$ is equivalent with $F_{large}$ against $P_{small}$. Combining these observations yields

$$\min_{f \in F_{small}} \max_{\mathbb{P} \in P_{large}} L(f, \mathbb{P}) = \min_{f \in F_{large}} \max_{\mathbb{P} \in P_{large}} L(f, \mathbb{P}) = \min_{f \in F_{large}} \max_{\mathbb{P} \in P_{small}} L(f, \mathbb{P}).$$

Therefore, in the worst case, there is no loss from considering random shuffling.

For general $K$, perform a random shuffle over all permutations $m$ of the labels $\{s_1, \cdots, s_K\}$. For any permutation $m$, define

$$f_m(a_1, \cdots, a_N) = m^{-1}\big(f\big(m(a_1), \cdots, m(a_N)\big)\big),$$

and draw $m$ uniformly at random; the aggregator then uses $f_m$ for aggregation. The same symmetrization argument applies as in the $K = 2$ case.

For general $K$, a symmetric condition is: for all $i$ and all $j \neq k$,

$$\mathbb{P}(A_i = s_j \mid S^* = s_j) = x_i, \qquad \mathbb{P}(A_i = s_k \mid S^* = s_j) = \frac{1 - x_i}{K - 1}.$$

We need to assume the original distribution has the symmetry property. This roughly says that each agent has the same accuracy for all questions, which might not hold in the real world, but is a reasonable approximation when we cannot estimate per-question accuracies and instead only have overall accuracies $x_i$ for each agent $i$. Importantly, this symmetry is only needed for the justification here. After random shuffling, it holds by construction.

## C EXTENSION: BEYOND CONDITIONAL INDEPENDENCE

A major reason that Assumption 1 may fail is the heterogeneity in question difficulty: when questions are easy, LLM agents all perform well, whereas their performance drops simultaneously when questions are difficult. This induces positive correlations across agents. In this section, we relax Assumption 1 and show that all of our theoretical results extend naturally to a more general setting.

Instead of assuming conditional independence among the whole dataset, we only assume conditional independence for a single question, which is quite natrual in real settings.

**Assumption 2.** *Given a single question, the outputs of agents are conditionally independent.*

Instead of using a single accuracy $x_i$ to summarize agent $i$'s performance, we introduce a question *difficulty level* $\alpha \in \mathbb{R}_{>0}$ and agent *ability* parameters $\beta_1, \ldots, \beta_N \in \mathbb{R}_{>0}$, inspired by settings in Whitehill et al. (2009); Li (2024). Here, a larger $\alpha$ indicates an easier question, and a larger $\beta$ indicates a stronger LLM agent. For each agent $i$ with ability $\beta_i$ and question difficulty $\alpha$, we assume that for true label $s^*$,

$$P(A_i = s^*) = \sigma_K(\alpha \beta_i) = \frac{e^{\alpha \beta_i}}{K - 1 + e^{\alpha \beta_i}} \text{ and } P(A_i = s) = \frac{1}{K - 1 + e^{\alpha \beta_i}} \text{ for any } s \neq s^*.$$

For $K = 2$, this reduces to the logistic function $P(A_i = S^*) = \frac{1}{1 + e^{-\alpha \beta_i}}$.

We use a difficulty distribution $D(\alpha)$ to quantify difficulty levels in the dataset, where $D(\alpha)$ can be any distribution over $\alpha \in \mathbb{R}_{>0}$. Let $\mathbb{P}_{D(\alpha)}$ denote the underlying joint distribution over the ground-truth label $S^*$ and all $N$ agents' outputs. Therefore, under Assumption 2

$$\mathbb{P}_{D(\alpha)}(A_1 = a_1, \cdots, A_N = a_N | S^* = s^*) = \mathbb{E}_{\alpha \sim D(\alpha)} \frac{\prod_{i: a_i = s^*} e^{\alpha \beta_i}}{\prod_{i=1}^N (K - 1 + e^{\alpha \beta_i})}.$$

Notice that when the dataset contains questions of varying difficulty, Assumption 1 no longer holds here. In other words, $\mathbb{P}(A_1, \ldots, A_N | S^*) \neq \prod_{i=1}^N \mathbb{P}(A_i | S^*)$ in the population (cf. Example 2). This observation breaks a fundamental assumption made in many works on information aggregation (Prelec et al., 2017; Arieli et al., 2018; Schoenebeck & Tao, 2021; Kong, 2024; Pan et al., 2024), thereby extending the theoretical frontier and offering new insights for the development of the aggregation literature.

**Example 2.** *Consider a case when $N = K = 2$, we assume $\alpha = 0$ with probability 0.3 and $\alpha = 1$ with probability 0.7. Then, it holds that in the population,*

$$\mathbb{P}(A_1, A_2 | S^*) \neq \mathbb{P}(A_1 | S^*) \mathbb{P}(A_2 | S^*).$$

*Proof of Example 2.* We now prove the statement in Example 2. We consider the probability in the population of the whole dataset, and we are going to show that even for each question, the output answers are independent, the conditional independence will be broken in the whole population. When $\alpha = 0$, every LLM agent has an accuracy of only 0.5, reflecting a hard question. On the other hand, for $\alpha = \infty$, the accuracy becomes 1, representing an easy question. In this extreme case, it holds that

$$\mathbb{P}(A_1 = s_1, A_2 = s_1 | S^* = s_1) = 0.3 * 0.5^2 + 0.7 = 0.775,$$

while

$$\mathbb{P}(A_1 = s_1, A_2 = s_2 | S^* = s_1) = \mathbb{P}(A_1 = s_2, A_2 = s_1 | S^* = s_1)$$
$$= \mathbb{P}(A_1 = s_2, A_2 = s_2 | S^* = s_1) = 0.075.$$

In the meantime, it holds that

$$\mathbb{P}(A_1 = s_1 | S^* = s_1) = \mathbb{P}(A_2 = s_1 | S^* = s_1) = 0.3 * 0.5 + 0.7 = 0.85.$$

Therefore, it holds that in the population,

$$\mathbb{P}(A_1 = s_1, A_2 = s_1 | S^* = s_1) = 0.775 \neq \mathbb{P}(A_1 = s_1 | S^* = s_1) \mathbb{P}(A_2 = s_1 | S^* = s_1) = 0.7225,$$

which ends the proof. $\square$

We can then naturally extend Algorithm 1 by setting the weight for agent $i$ directly to $\beta_i$, see the details in Algorithm 3. Notably, this extension does not require any knowledge of $D(\alpha)$. The Bayesian optimality of Algorithm 3 still holds.

**Theorem 4.** *Under Assumption 2, for any $D(\alpha)$, $f_{EOW}$ defined in Algorithm 3 is the Bayesian optimal aggregator.*

For second-order information, we prove that even without knowing $\alpha$, Algorithm 2 still outperforms majority voting. More formally, we establish the following theorem.

**Theorem 5.** *Under Assumption 2, for any $D(\alpha)$, ISP in Algorithm 2 outperforms MV, which in turn outperforms SP, in expectation. That is, $\mathbb{E}[Adv_{ISP}(s^*)] \geq \mathbb{E}[Adv_{MV}(s^*)] \geq \mathbb{E}[Adv_{SP}(s^*)]$.*

The proofs of Theorems 4 and 5 are delayed to Appendix G.

---

**Algorithm 3** Extended Optimal Weight (EOW) Algorithm

---

1: **Input:** Agent ability parameters $\beta_1, ..., \beta_N$.
2: Pre-processing: Randomly shuffle candidate labels, obtaining the shuffle mapping $\pi$.
3: Observe predictions $a_1, ..., a_N$.
4: Aggregate via $f_{EOW}(a_1, \cdots, a_N) = \arg\max_{s \in S} \sum_{i=1}^{N} \beta_i \mathbb{1}\{a_i = s\}$.
5: **Output:** Map back using $\pi^{-1}$ and return $\pi^{-1}(f_{EOW}(a_1, \cdots, a_N))$.

---

# D    OMITTED DETAILS IN SECTION 3

## D.1    PROOF OF THEOREM 1

We first design a bijection between accuracies and a new vector $\beta$ which represents agents' ability. We assume $x_i = \sigma_K(\beta_i) = \frac{e^{\beta_i}}{K - 1 + e^{\beta_i}}$ for any $x_i$. When the accuracy $x_i$ increases, the agent's corresponding ability $\beta_i$ will be up as well. Then, due to the random shuffle, it's equivalent to assume that $\mathbb{P}(A_i = S^*) = \frac{e^{\beta_i}}{K - 1 + e^{\beta_i}}$ and $\mathbb{P}(A_i \neq S^*) = \frac{K-1}{K - 1 + e^{\beta_i}}$.

It then holds that

$$
\mathbb{P}(S^* = s_i | A_1 = a_1, ..., A_N = a_N) = \frac{\mathbb{P}(A_1 = a_1, ..., A_N = a_N | S^* = s_i)\mathbb{P}(S^* = s_i)}{\mathbb{P}(A_1 = a_1, ..., A_N = a_N)}
$$

$$
\propto \mathbb{P}(A_1 = a_1, ..., A_N = a_N | S^* = s_i)
$$

$$
= \prod_{j=1}^{N} \mathbb{P}(A_j = a_j | S^* = s_i)
$$

$$
= \prod_{j=1}^{N} [\frac{e^{\beta_j}}{K - 1 + e^{\beta_j}} \mathbb{1}\{a_j = s_i\} + \frac{1}{K - 1 + e^{\beta_j}} \mathbb{1}\{a_j \neq s_i\}]
$$

$$
= \prod_{j=1}^{N} \frac{e^{\beta_j \cdot \mathbb{1}\{a_j = s_i\} + 0 \cdot \mathbb{1}\{a_j \neq s_i\}}}{K - 1 + e^{\beta_j}}
$$

$$
= \frac{e^{\sum_{\{j : a_j = s_i\}} \beta_j}}{\prod_{j=1}^{N}(K - 1 + e^{\beta_j})}
$$

$$
\propto e^{\sum_{\{j : a_j = s_i\}} \beta_j}.
$$

The second formula holds as $\mathbb{P}(S^* = s_i) = \frac{1}{K}$ and $\mathbb{P}(A_1, ..., A_N)$ is independent of $s_i$. The second equation holds due to Assumption 1, and we have the third equation as agents are homogeneous with respect to incorrect options, say $\mathbb{P}(A_i = s_i) = \frac{1}{K - 1 + e^{\beta_i}}$ for every $s_i \neq s^*$.

Since the Bayesian optimal aggregator chooses $\hat{s} = \arg\max_s \mathbb{P}(S^* = s | A_1 = a_1, ..., A_N = a_N)$, we only need to compare among $\sum_{\{i : a_i = s\}} \beta_i$ for all labels. It's equivalent to aggregate as

$$
\hat{s} = \arg\max_s \sum_{\{i : a_i = s\}} \beta_i = \arg\max_x \sum_{\{i : a_i = s\}} \sigma_K^{-1}(x_i),
$$

which finishes our proof.

## D.2    PROOF OF COROLLARY 1

When $K = 2$, from the proof of Theorem 1, we immediately know that the optimal weight is $\omega_i = \sigma_2^{-1}(x_i)$. Note that $\sigma_2(x) = \frac{e^x}{1 + e^x}$ which is exactly the logistic function $\sigma(x)$. It yields that the optimal weights are proportional to the logits of the accuracies, and closes off the proof.

## D.3    PROOF OF COROLLARY 2

Recall that Theorem 1 proves that the optimal weights are proportional to the logits of the accuracies. When all agents are homogeneous, say $x_1 = x_2 = \cdots = x_N$, every weight becomes the same.

Since the result of $\arg\max$ is invariant to scaling, it's equivalent to set all weights to 1, aligned with majority voting. Therefore, we prove that majority voting is the optimal strategy for homogeneous agents, justifying the use of majority voting for reasoning with a single model.

### D.4 PROOF OF PROPOSITION 2

From Theorem 1, we already show that $f_{OW}$ is Bayesian optimal. Therefore, when comparing $f_{OW}$ and $f_i(a_1, \cdots, a_N) = a_i$, we only need to figure out whether $f_i$ is Bayesian optimal.

From Appendix D.1, it holds that

$$\mathbb{P}(S^* = s \mid A_1 = a_1, \cdots, A_N = a_N) = \frac{\mathbb{P}(S^* = s)}{\mathbb{P}(A_1 = a_1, \cdots, A_N = a_N)} \times \frac{e^{\sum_{\{j:a_j=s\}} \beta_j}}{\prod_{j=1}^{N}(K - 1 + e^{\beta_j})}, \tag{8}$$

where $\beta_i = \sigma_K^{-1}(x_i)$ for all $i \in [K]$.

If $f_i$ is Bayesian optimal, for any $a_1, \cdots, a_N, s$, it holds that

$$P(S^* = a_i \mid A_1 = a_1, \cdots, A_N = a_N) \geq P(S^* = s \mid A_1 = a_1, \cdots, A_N = a_N).$$

Taking back to Equation (8), we obtain

$$\frac{\mathbb{P}(S^* = a_i)}{\prod_{j=1}^{N}(K - 1 + e^{\beta_j})} e^{\sum_{\{j:a_j=a_i\}} \beta_j} \geq \frac{\mathbb{P}(S^* = s)}{\prod_{j=1}^{N}(K - 1 + e^{\beta_j})} e^{\sum_{\{j:a_j=s\}} \beta_j}.$$

Because $\mathbb{P}(S^* = s) = \frac{1}{K}$, it is then equivalent to

$$\sum_{\{j:a_j=a_i\}} \beta_j \geq \sum_{\{j:a_j=s\}} \beta_j$$

for any $a_1, \cdots, a_N, s$.

Let $s$ be any label except $a_i$, and $a_j = s \neq a_i$ for all $j \neq i$. It holds that

$$\beta_i \geq \sum_{j \neq i} \beta_i,$$

which concludes the proof.

## E OMITTED DETAILS IN SECTION 4

### E.1 BASIC PROPERTIES OF SECOND-ORDER INFORMATION

**Proposition 3.** *After random shuffling, the second-order information has the following propositions.*

- *Exchangability:* $\mathbb{P}(A_i|A_j) = \mathbb{P}(A_j|A_i)$ *for every* $(i, j)$;

- *Symmetry:* $\mathbb{P}(A_i = s_k|A_j = s_k) = \mathbb{P}(A_i = s_l|A_j = s_l)$ *for every* $(i, j, k, l)$;

- *Monotonicity: If* $\widetilde{x}_i \geq x_i$, *it holds that* $\mathbb{P}(\widetilde{A}_i = s_k|A_j = s_k) \geq \mathbb{P}(A_i = s_k|A_j = s_k)$ *for every* $(i, j, k)$;

- *Null information: If* $x_i = \frac{1}{K}$, *it holds that* $\mathbb{P}(A_i = s_k|A_j = s_l) = \frac{1}{K}$ *for every* $(i, j, k, l)$.

*Proof of Proposition 3.* From Proposition 1, we have $\mathbb{P}(S^* = s_i) = \frac{1}{K}$, $\mathbb{P}(A_i = s_j|S^* = s_j) = x_i$ and $\mathbb{P}(A_i = s_k|S^* = s_j) = \frac{1-x_i}{K-1}$ for every $s_k \neq s_j$.

For exchangability, it holds that if $a_i = a_j$, we'd have

$$\mathbb{P}(A_i = a_i|A_j = a_j) = \mathbb{P}(A_j = a_j|A_i = a_i) = x_i x_j + \frac{(1 - x_i)(1 - x_j)}{K - 1}.$$

If $a_i \neq a_j$, it holds that

$$\mathbb{P}(A_i = a_i | A_j = a_j) = \mathbb{P}(A_j = a_j | A_i = a_i) = \frac{x_i(1 - x_j) + (1 - x_i)x_j}{K - 1} + \frac{(K - 2)(1 - x_i)(1 - x_j)}{(K - 1)^2}.$$

Hence, it holds that $\mathbb{P}(A_i = a_i | A_j = a_j) = \mathbb{P}(A_j = a_j | A_i = a_i)$.

For symmetry, it holds that

$$\mathbb{P}(A_i = s_k | A_j = s_k) = x_i x_j + \frac{(1 - x_i)(1 - x_j)}{K - 1}.$$

Note that this probability is independent of label $s_k$; therefore, we prove the property of symmetry.

For monotonicity, we know that when agent $i$ has accuracy $x_i$, the required probability is

$$\mathbb{P}(A_i = s_k | A_j = s_k) = x_i x_j + \frac{(1 - x_i)(1 - x_j)}{K - 1}.$$

It then holds that

$$\frac{\partial \mathbb{P}(A_i = s_k | A_j = s_k)}{\partial x_i} = x_j - \frac{1 - x_j}{K - 1} \geq 0.$$

The inequality holds because $x_j \geq \frac{1}{K}$. Therefore, the monotonicity holds automatically.

For null information, there are two different cases. When $s_k = s_l$, it holds that

$$\mathbb{P}(A_i = s_k | A_j = s_l) = x_i x_j + \frac{(1 - x_i)(1 - x_j)}{K - 1} = \frac{x_j}{K} + \frac{1 - x_j}{K} = \frac{1}{K}.$$

On the other hand, when $s_k \neq s_l$, it holds that due to exchangability,

$$\mathbb{P}(A_i = s_k | A_j = s_l) = \frac{x_i(1 - x_j) + (1 - x_i)x_j}{K - 1} + \frac{(K - 2)(1 - x_i)(1 - x_j)}{(K - 1)^2} = \frac{1}{K}.$$

It then finishes our proof. $\qquad\square$

### E.2 PROOF OF EXAMPLE 1

With some calculations, we notice that in the first 3 cases, the advantage of label $s_1$ is larger compared to $s_2$ for both $SP$ and $ISP$. However, for the last case that $(A_1, A_2, A_3, A_4) = (s_1, s_1, s_2, s_2)$, SP's advantage of $s_1$ is $-\frac{1}{3}$, so SP will output $s_2$. On the other hand, ISP's advantage of $s_1$ is $\frac{1}{3}$. Therefore, ISP will output $s_1$ correctly. As MV breaks the tie uniformly, with probability $\frac{1}{2}$, it will output correctly in case 4, while with another probability of $\frac{1}{2}$, the prediction will be wrong. Therefore, we know that the accuracies are $\frac{7}{8}$, $\frac{3}{4}$ and 1 for MV, SP and ISP, respectively.

We now give another example with an odd number of agents.

**Example 3.** *We assume there are 9 agents with accuracies $x_1 = x_2 = x_3 = x_4 = 1$ and $x_5 = x_6 = x_7 = x_8 = x_9 = 0.5$ with $K = 2$ candidates. We assume the true label after shuffling is $s_1$ without loss of generality because of symmetry. Since there are odd agents, there is no tie. Only in the case when $A_5 = A_6 = A_7 = A_8 = A_9 = s_2$, MV will get a wrong aggregation with probability $\frac{1}{32}$. For SP, when there are either 4 or 5 $s_1$, the total score is the same as $\sum_{i=1}^{9} S_{SP}(s_1, i) = \frac{21}{4} > 5$. Therefore, the probability of giving a wrong prediction is $\frac{3}{16}$. For ISP, however, the total score is only $\frac{15}{4} < 4$. Hence, ISP will output the correct prediction with probability 1 as there are at least 4 $s_1$.*

### E.3 PROOF OF THEOREM 2

After random shuffling, we know that the prior over labels is a uniform distribution. Notice that $Adv_{MV}(s^*) = \sum_i \mathbb{1}\{a_i = s^*\} - \frac{N}{K}$. It holds that

$$\mathbb{E}[Adv_{MV}(s^*)] = \sum_i x_i - \frac{N}{K} = \sum_i (x_i - \frac{1}{K}).$$

Let's consider the first agent with accuracy $x_1$, its contribution to $\mathbb{E}[Adv_{MV}(s^*)]$ is $x_1 - \frac{1}{K} := q_1$. Similarly, we can rephrase $Adv_{ISP}(s^*)$. We first assume there are only two agents. When there are more than two agents, we need to take the average over the agents' predictions that we condition on. It holds with some calculations that

$$\mathbb{P}(A_1 = s^*|A_2 = s^*) = x_1 x_2 + \frac{(1-x_1)(1-x_2)}{K-1},$$

and for any $s \neq s^*$,

$$\mathbb{P}(A_1 = s^*|A_2 = s) = \frac{1}{K-1}x_1(1-x_2) + \frac{1}{K-1}(1-x_1)x_2 + \frac{K-2}{(K-1)^2}(1-x_1)(1-x_2).$$

Since for all $s \neq s^*$ we have the same conditional probability, we use $\mathbb{P}(A_1 = s^*|A_1 = \neg s^*)$ to represent this probability with a little abuse of notations.

Therefore, the first agent has a contribution $q_2$ of

$$x_1 - x_2\mathbb{P}(A_1 = s^*|A_2 = \neg s^*) - (1-x_2)[\frac{1}{K-1}\mathbb{P}(A_1 = s^*|A_2 = s^*) + \frac{K-2}{K-1}\mathbb{P}(A_1 = s^*|A_2 = \neg s^*)].$$

With probability $x_2$, the second agent will predict as $s^*$. So, the corresponding conditional probability is $\mathbb{P}(A_1 = s^*|A_2 = \neg s^*)$. On the other hand, with probability $1 - x_2$, $x_2$ will output another wrong answer. Then, within the $K - 1$ possible exploration events, one is $\mathbb{P}(A_1 = s^*|A_2 = s^*)$ while the other $K - 2$ are $\mathbb{P}(A_1 = s^*|A_2 = \neg s^*)$.

We then calculate the gap between $q_1$ and $q_2$. We notice that either $x_1 = \frac{1}{K}$ or $x_2 = \frac{1}{K}$ will lead to zero gap due to Proposition 3. It then holds that

$$q_2 - q_1 = \frac{1}{(K-1)^3}(x_1 - \frac{1}{K})(Kx_2 - 1)^2.$$

Therefore, when there are more than two agents, the weight of each agent in the contribution $q_2$ is $\frac{1}{N-1}$, which yields that the difference between agent 1's contributions to ISP and MV is now

$$\sum_{i=2}^{N} \frac{1}{N-1}\frac{1}{(K-1)^3}(x_1 - \frac{1}{K})(Kx_i - 1)^2 = \sum_{i=2}^{N} \frac{1}{N-1}\frac{1}{K(K-1)^3}(Kx_1 - 1)(Kx_i - 1)^2.$$

Recall that the total advantage is the sum of every agent's contribution. It then holds that

$$\mathbb{E}[Adv_{ISP}(s^*) - Adv_{MV}(s^*)] = \frac{\sum_{i=1}^{N}\sum_{j\in[N]\setminus\{i\}}(Kx_i - 1)(Kx_j - 1)^2}{(N-1)K(K-1)^3},$$

due to the linearity of the expectation.

For any fixed accuracies $x_1, ..., x_N$, we notice that the numerator is proportional to $\Theta(N^2K^3)$ and the denominator is roughly $\Theta(NK^4)$. Therefore, we know that $\mathbb{E}[Adv_{ISP}(s^*) - Adv_{MV}(s^*)] \approx \Theta(\frac{N}{K})$. Since the advantage is the sum of every agent's contribution, it's reasonable to be proportional to $N$. More notably, the advantage of ISP over MV is proportional to $\frac{1}{K}$. In other words, as the number of options $K$ increases, the performance gap between them diminishes. This is because ISP motivates agents to reflect on alternative options in order to explore the unknown environment. However, as $K$ grows, since we only conduct a single round of uniform reflection, i.e., computing the conditional probability given $\neg A_i$, the probability of reaching the correct answer is roughly $\frac{1}{K}$ if the original prediction is wrong. As a result, the benefit of ISP relative to MV is most significant in the region of a few options. For tasks with many possible options or open-ended problems, such as text generation, we may need to design additional reflection or exploration mechanisms, which we leave for future work.

For SP, we also presume there are $N = 2$ agents. So, the contribution of the first agent is now

$$x_1 - x_2\mathbb{P}(A_1 = s^*|A_2 = s^*) - (1-x_2)\mathbb{P}(A_1 = s^*|A_2 = \neg s^*) := q_3.$$

This is because agent 2 will predict $s^*$ with probability $x_2$ and answer mistakenly with another probability $1 - x_2$. Hence, we only need to compute the gap between $q_1$ and $q_3$ that

$$q_1 - q_3 = \frac{(Kx_1 - 1)(Kx_2 - 1)^2}{K(K-1)^2}.$$

When there are more than two agents, the gap would become

$$\frac{1}{K} - S_{SP}(s^*, 1) = \frac{\sum_{i=2}^{N}(Kx_1 - 1)(Kx_i - 1)^2}{(N-1)K(K-1)^2}.$$

Therefore, adding the difference of all agents yields

$$\mathbb{E}[Adv_{MV}(s^*) - Adv_{SP}(s^*)] = \frac{\sum_{i=1}^{N}\sum_{j\in[N]\setminus\{i\}}(Kx_i - 1)(Kx_j - 1)^2}{(N-1)K(K-1)^2}.$$

Unlike the gap between ISP and MV, this gap is $\Theta(N)$ with increasing option number $K$ for any fixed $x_1, ..., x_N$. This shows that SP is consistently weaker than MV by a constant factor. The reason is that when each agent answers incorrectly with some fixed probability, it tends to get stuck on the wrong option, rather than reflecting on how other agents would behave under alternative predictions as ISP does. As a result, its performance can be even worse than simple majority voting. This demonstrates that the surprising popularity mechanism, originally designed for incentive settings, is not suitable for LLM aggregation scenarios, where agents have no incentive to deliberately provide incorrect answers.

We further observe that as model accuracy $x_i$ improves, both gaps increase accordingly. Nevertheless, because higher model accuracy inherently elevates the likelihood that all three methods converge to the correct prediction, the relative improvement of ISP over MV may exhibit a more nuanced and non-monotonic behavior. We leave a deeper characterization of this phenomenon to future work.

Finally, since we know that every agent is better than a random predictor, say $x_i \geq \frac{1}{K}$, it holds that

$$\mathbb{E}[Adv_{ISP}(s^*)] \geq \mathbb{E}[Adv_{MV}(s^*)] \geq \mathbb{E}[Adv_{SP}(s^*)],$$

which ends the proof. Note that, due to symmetry, whenever $s^*$ has a larger advantage, other labels $s \in \neg s^*$ must necessarily have smaller advantages. Therefore, compared to MV and SP, ISP is more likely to identify that label $s^*$ possesses the greater advantage, and thus aggregate toward the correct prediction of the ground truth label.

### E.4 PROOF OF THEOREM 3

We notice that after a random shuffle, the expectation of the advantage of ISP equals

$$\mathbb{E}[Adv_{ISP}(s^*)] = \sum_{i=1}^{N}\left[x_i - \frac{1}{(N-1)(K-1)}\sum_{j\in[N]\setminus\{i\}}\mathbb{E}[\sum_s \mathbb{1}\{a_j = s\}\sum_{\widetilde{s}\in\neg s}\mathbb{P}(A_i = s^*|A_j = \widetilde{s})]\right],$$

due to the linearity of expectation. Therefore, we only need to bound the distance between $\mathbb{E}[\mathbb{1}\{a_j = s\}\widehat{\mathbb{P}}(A_i = s^*|A_j = \widetilde{s})]$ and $\mathbb{P}(A_j = s|S^* = s^*)\mathbb{P}(A_i = s^*|A_j = \widetilde{s})$ for any pair $(s, \widetilde{s})$.

We know that $\mathbb{E}[\mathbb{1}\{a_j = s\}] = \mathbb{P}(A_j = s|S^* = s^*)$ when the true label is $s^*$, so the only issue is that $\mathbb{1}\{a_j = s\}$ and $\widehat{\mathbb{P}}(A_i = s^*|A_j = \widetilde{s})$ are correlated. Assuming we have $M$ i.i.d. questions, we now consider the aggregation for the $m$-th question, and we use the subscript $-m$ to represent all questions except the $m$-th one. $\widehat{\mathbb{P}}_{-m}$ means that we exclude the $m$-th question when computing the empirical conditional probability. Therefore, it holds that

$$|\mathbb{E}[\mathbb{1}\{a_j = s\}\widehat{\mathbb{P}}(A_i = s^*|A_j = \widetilde{s})] - \mathbb{P}(A_j = s|S^* = s^*)\mathbb{P}(A_i = s^*|A_j = \widetilde{s})|$$

$$\leq|\mathbb{E}[\mathbb{1}\{a_j = s\}\widehat{\mathbb{P}}(A_i = s^*|A_j = \widetilde{s})] - \mathbb{E}[\mathbb{1}\{a_j = s\}\widehat{\mathbb{P}}_{-m}(A_i = s^*|A_j = \widetilde{s})]|$$

$$+ |\mathbb{E}[\mathbb{1}\{a_j = s\}\widehat{\mathbb{P}}_{-m}(A_i = s^*|A_j = \widetilde{s})] - \mathbb{P}(A_j = s|S^* = s^*)\mathbb{P}(A_i = s^*|A_j = \widetilde{s})|$$

$$\leq|\widehat{\mathbb{P}}(A_i = s^*|A_j = \widetilde{s}) - \widehat{\mathbb{P}}_{-m}(A_i = s^*|A_j = \widetilde{s})|$$

$$+ |\mathbb{P}(A_j = s|S^* = s^*)\widehat{\mathbb{P}}_{-m}(A_i = s^*|A_j = \widetilde{s}) - \mathbb{P}(A_j = s|s^*)\mathbb{P}(A_i = s^*|A_j = \widetilde{s})|$$

$$\leq|\widehat{\mathbb{P}}(A_i = s^*|A_j = \widetilde{s}) - \widehat{\mathbb{P}}_{-m}(A_i = s^*|A_j = \widetilde{s})| + |\widehat{\mathbb{P}}_{-m}(A_i = s^*|A_j = \widetilde{s}) - \mathbb{P}(A_i = s^*|A_j = \widetilde{s})|.$$

The first inequality holds due to the triangle inequality, while the second one holds as $0 \leq \mathbb{1}\{a_j = s\} \leq 1$ and $A_j$ is independent of $\widehat{\mathbb{P}}_{-m}$. The third inequality holds due to $\mathbb{P}(A_j = s|S^* = s^*) \leq 1$.

For the term $|\widehat{\mathbb{P}}(A_i = s^*|A_j = \widetilde{s}) - \widehat{\mathbb{P}}_{-m}(A_i = s^*|A_j = \widetilde{s})|$, recall that $\widehat{\mathbb{P}}(A_i = s^*|A_j = \widetilde{s}) = \frac{\#\{A_i = s^*, A_j = \widetilde{s}\}}{\#\{A_j = \widetilde{s}\}}$ and $\widehat{\mathbb{P}}_{-m}(A_i = s^*|A_j = \widetilde{s}) = \frac{\#_{-m}\{A_i = s^*, A_j = \widetilde{s}\}}{\#_{-m}\{A_j = \widetilde{s}\}}$. Using Hoeffding's inequality (Wainwright, 2019), it holds that with probability at least $1 - \mathcal{O}(\delta)$,

$$|\#\{A_i = s^*, A_j = \widetilde{s}\} - M\mathbb{P}(A_i = s^*, A_j = \widetilde{s})| \lesssim \mathcal{O}(\sqrt{M \log(\frac{1}{\delta})}), \qquad (9)$$

and

$$|\#\{A_j = \widetilde{s}\} - M\mathbb{P}(A_j = \widetilde{s})| \lesssim \mathcal{O}(\sqrt{M \log(\frac{1}{\delta})}). \qquad (10)$$

In addition, we know that $|\#\{A_i = s^*, A_j = \widetilde{s}\} - \#_{-m}\{A_i = s^*, A_j = \widetilde{s}\}| \leq 1$ and $|\#\{A_j = \widetilde{s}\} - \#_{-m}\{A_j = \widetilde{s}\}| \leq 1$. It then holds that

$$|\widehat{\mathbb{P}}(A_i = s^*|A_j = \widetilde{s}) - \widehat{\mathbb{P}}_{-m}(A_i = s^*|A_j = \widetilde{s})| \lesssim \mathcal{O}(\frac{1}{M}),$$

when $M \gtrsim \Omega(\log(\frac{1}{\delta}))$.

For the second term $|\widehat{\mathbb{P}}_{-m}(A_i = s^*|A_j = \widetilde{s}) - \mathbb{P}(A_i = s^*|A_j = \widetilde{s})|$, by replacing $\#\{A_i = s^*, A_j = \widetilde{s}\}$ and $\#\{A_j = \widetilde{s}\}$ with $\#_{-m}\{A_i = s^*, A_j = \widetilde{s}\}$ and $\#_{-m}\{A_j = \widetilde{s}\}$ respectively in Inequalities 9 and 10, it then holds that

$$|\widehat{\mathbb{P}}_{-m}(A_i = s^*|A_j = \widetilde{s}) - \mathbb{P}(A_i = s^*|A_j = \widetilde{s})| \lesssim \mathcal{O}(\sqrt{\frac{1}{M} \log(\frac{1}{\delta})}),$$

when $M$ is sufficiently large.

Therefore, we know that with probability at least $1 - \Theta(\delta)$, for every question, it holds that

$$|\mathbb{E}[\widehat{Adv}_{ISP}(s^*)] - \mathbb{E}[Adv_{ISP}(s^*)]| \lesssim \widetilde{\mathcal{O}}(\sqrt{\frac{1}{M} \log(\frac{1}{\delta})}).$$

Here, we have $M$ questions in total, so the cumulative failure probability is polynomial in $M$. Hence, we use $\widetilde{\mathcal{O}}(\cdot)$ to hide some logarithmic terms of $M$ in the concentration upper bound.

Together with Theorem 2 and choosing the constant before the failure probability $\delta$ elaborately, it yields that

$$\mathbb{E}[\widehat{Adv}_{ISP}(s^*) - Adv_{MV}(s^*)] \geq \mathbb{E}[Adv_{ISP}(s^*) - Adv_{MV}(s^*)] - |\mathbb{E}[\widehat{Adv}_{ISP}(s^*) - Adv_{ISP}(s^*)]|$$

$$\gtrsim \frac{\sum_{i=1}^{N} \sum_{j \in [N] \setminus \{i\}} (Kx_i - 1)(Kx_j - 1)^2}{(N-1)K(K-1)^3} - \widetilde{\mathcal{O}}(\sqrt{\frac{1}{M} \log(\frac{1}{\delta})}),$$

which ends the proof.

# F OMITTED DETAILS IN SECTION 5

## F.1 OMITTED DETAILS IN SECTION 5.1

For the implementation of MV, we break ties randomly, and this procedure has little effect on the overall results. In Figure 1, we observe that as the number of choices $K$ increases, the gap between ISP and MV gradually decreases, validating Theorem 2. At the same time, the gap between MV and SP first decreases and then stabilizes. This initial reduction arises because, as $K$ grows, the accuracies of all three algorithms improve, thereby reducing the potential for further gains. Decomposing the factors that contribute to the decrease is an interesting direction for future research.

## F.2 OMITTED DETAILS IN SECTION 5.2

With some calculations, it holds that

$$\mathbb{P}(A_i = s_k|A_j = s_l)[x_1, ..., x_N] = \begin{cases} x_i x_j + \frac{(1-x_i)(1-x_j)}{K-1}, & \text{if } s_k = s_l, \\ \frac{x_i(1-x_j)+(1-x_i)x_j}{K-1} + \frac{(K-2)(1-x_i)(1-x_j)}{(K-1)^2}, & \text{if } s_k \neq s_l. \end{cases}$$

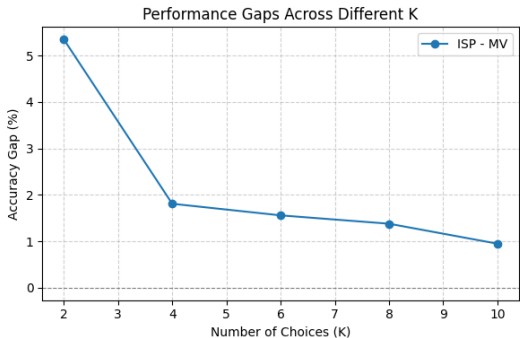

Figure 1: The performance gap between ISP and MV vanishes as $K$ increases.

Together with $\widehat{\mathbb{P}}(A_i = s_k | A_j = s_l) = \frac{\#\{A_i = s_k, A_j = s_l\}}{\#\{A_j = s_l\}}$, we can use some gradient descent method to solve Equation (7), which yields

$$\widehat{x}_1, ..., \widehat{x}_N = \underset{x_1,...,x_N}{\arg\min}\Big[ \sum_{i,j,k}(x_i x_j + \frac{(1-x_i)(1-x_j)}{K-1} - \frac{\#\{A_i = s_k, A_j = s_k\}}{\#\{A_j = s_k\}})^2$$

$$+ \sum_{i,j,k,l:k\neq l}(\frac{x_i(1-x_j) + (1-x_i)x_j}{K-1} + \frac{(K-2)(1-x_i)(1-x_j)}{(K-1)^2} - \frac{\#\{A_i = s_k, A_j = s_l\}}{\#\{A_j = s_l\}})^2\Big].$$

We can also explore alternative loss functions rather than squared loss here, which we leave as a direction for future research.

### F.3 OMITTED DETAILS IN SECTION 5.3

For the GPT family, we access the models exclusively through the Microsoft Azure API, using the default template with temperature=1.0 and top_p=1.0. For the other open-source models, we generate responses on a single A100 GPU using vLLM (Kwon et al., 2023) with default seed 0.

The UltraFeedback dataset contains 63,135 questions. We retain only those that comply with OpenAI's content management policy, with 56,380 left. For each retained question, we randomly shuffle the two candidate responses and query different LLM agents using the following prompt template. We instruct the LLMs to provide only the final answer without any explanation, which facilitates cleaner data processing. While prompt engineering may further improve the accuracy of individual model responses, we leave this as an interesting direction for future research.

```
**UltraFeedback prompt**: You are a helpful assistant.

Original question, e.g., "Prompt: how can i develop a habit of
drawing daily".

There are two answers:
A. One response, e.g., "Developing a daily habit of drawing can
be challenging but with consistent practice and a few tips, it
can become an enjoyable and rewarding part of your daily routine.
Here are some strategies to help you develop the habit of drawing
daily..."

B. The other response, e.g., "As an AI language model, I cannot
personally develop habits for you. But, here are some tips for
developing a habit of drawing daily..."

Which answer is better? Reply with only 'A' or 'B'. Do not explain
or repeat the answers.
```

The MMLU dataset contains 115,700 questions. Among them, 109,820 questions are successfully answered by all LLMs. For each question, we first randomly shuffle the four answer choices. We then query the models using the following prompt template.

**MMLU prompt**: You are a helpful assistant.

Original question, e.g., "A little stress is good, since it helps you keep motivated to meet your goals. However, too much stress is bad for your health...According to the passage, which of the following is NOT true?"

There are four options.
A. One option, e.g., "It's unnecessary for you to do all the tasks by yourself."

B. One option, e.g., "It helps you stay motivated to think about the undone tasks."

C. One option, e.g., "It helps if you put your attention to one task at a time."

D. One option, e.g., "It's better to plan a specific time to do the task."

Which option is correct? Reply with only 'A', 'B', 'C', or 'D'. Do not explain or repeat the answer."

The ARMMAN dataset contains health information and call records for 12,000 women. 11,785 cases are successfully answered by all LLMs. We treat a call as valid if its duration exceeds ten seconds. We provide the LLMs with the call reception records from the past ten weeks, and then query the models using the following simple prompt template.

**ARMMAN prompt**: You are a helpful assistant.

Brief introduction, e.g., "You are a mother enrolled in the ARMMAN Maternal and Child Healthcare Mobile Health program. ARMMAN is a nongovernmental organization in India dedicated to reducing maternal and neonatal mortality among underprivileged communities..."

Your Background: e.g., "You enrolled in the program during the 9 week of your pregnancy. You are 35+ years old...You have had 0 stillbirth(s) and have 2 living child(ren)."

Past Behavior: e.g., "The following is a record of your previous listening behavior (each representing one week): – Week 1: Listened for 0 seconds, Receive an automated voice message...– Week 10: Listened for 0 seconds, Receive an automated voice message"

This Week's Call Type: e.g., "You will receive an automated voice message this week."

Task, e.g., "Based on this information, as well as the context of the program and on typical behavior of mothers in India, decide whether you will be engaged with the next health message..."

Question: e.g., "Will you be engaged with the next health message (listening for at least 10 seconds)?"

```
Instruction, e.g., "Please decide whether you will be engaged with
the next health message: '##Yes##' for engagement (listening at
least 10 seconds) or '##No##' for lack of engagement (listening
less than 10 seconds) in this week."

There are two answers:
A. One option, e.g., "##Yes##"

B. The other option, e.g., "##No##"

Which answer is better? Reply with only 'A' or 'B'. Do not explain
or repeat the answers."
```

Beyond the generation, all our training-free aggregation steps can be conducted on a personal CPU with negligible cost. For instance, executing locally with an Apple M3 Pro CPU, the accuracy-estimation step of OW-L and OW-I on UltraFeedback takes only 10.3 and 6.1 seconds, respectively. The second aggregation step only needs around 0.1 seconds. Since the dominant cost in modern LLM pipelines typically comes from gradient-based training, our training-free approach avoids this bottleneck entirely. Enhancing the reasoning and aggregation stage in this way allows individuals and small organizations, who may lack large compute resources but can easily perform inference (e.g., calling APIs), to benefit from LLM improvements and contribute more effectively to the community.

Moreover, we find that the distribution of answers from the selected models is nearly uniform over the candidates, for example, on the ARMMAN Healthcare dataset, GPT-4o-2024-11-20 selects option $A$ 49.7% of the time and option $B$ 50.3% while Qwen2.5-Instruct- 14B chooses $A$ 51.4% and $B$ 48.6%, indicating minimal position effects. Therefore, we conclude that random shuffling does not affect the model responses significantly.

### F.4 OMITTED DETAILS IN SECTION 5.4

Within each family, we use "S" to denote the strong model and "W" to denote the weak model. Accordingly, we abbreviate GPT-4o-2024-11-20, GPT-35-turbo-0125, Qwen2.5-Instruct-14B, Qwen2.5-Instruct-3B, Llama3.1-8B-Instruct, Llama3.2-1B-Instruct, Phi-4 and Phi-4-mini-instruct as GS, GW, QS, QW, LS, LW, PS and PW, respectively. We begin by presenting in Figure 2 the overall performance of these eight models on UltraFeedback, MMLU, and ARMMAN.

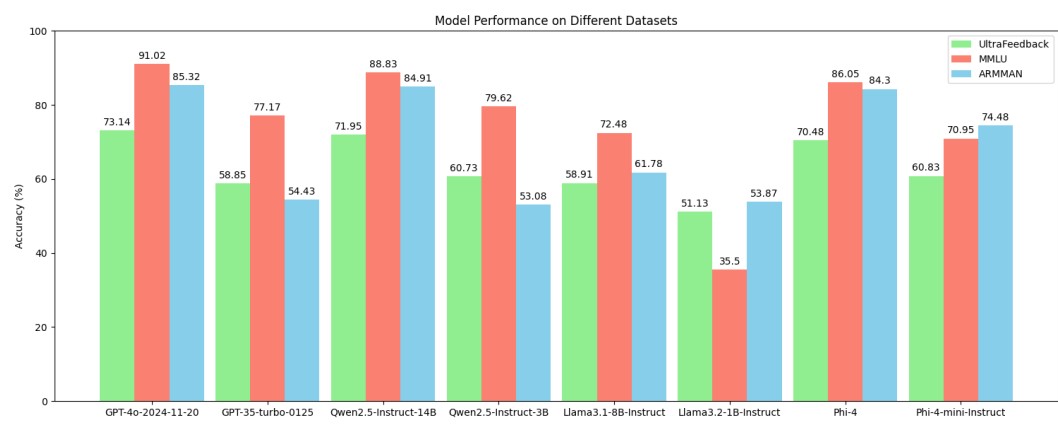

Figure 2: Accuracy of different LLMs across the three datasets.

We now present the results of the 16 ensembles on UltraFeedback in Table 5. We observe that MV outperforms ISP only in the case of aggregating GPT-4o-2024-11-20, Qwen2.5-Instruct-3B,

Llama3.2-1B-Instruct and Phi-4. In all other cases, our three algorithms achieve higher accuracy than MV.

Table 5: Experimental Results: We report the overall accuracy for different LLM ensembles on UltraFeedback. In the tables, the best results are highlighted in **bold**.

| Accuracy | OW-L | OW-I | ISP | MV |
|---|---|---|---|---|
| (GS, QS, LS, PS) | **73.66%** | **73.66%** | 73.26% | 72.21% |
| (GS, QS, LS, PW) | **72.64%** | **72.64%** | 72.09% | 69.89% |
| (GW, QS, LS, PS) | 70.88% | **71.15%** | 70.78% | 68.01% |
| (GW, QS, LS, PW) | **69.00%** | **69.00%** | 67.73% | 65.11% |
| (GS, QS, LW, PS) | **73.66%** | **73.66%** | 73.18% | 72.11% |
| (GS, QS, LW, PW) | **72.64%** | **72.64%** | 71.57% | 69.23% |
| (GW, QS, LW, PS) | **71.15%** | **71.15%** | 70.46% | 67.20% |
| (GW, QS, LW, PW) | **67.10%** | **67.10%** | 66.77% | 64.18% |
| (GS, QW, LS, PS) | 71.19% | 71.19% | **71.74%** | 71.15% |
| (GS, QW, LS, PW) | **69.17%** | **69.17%** | 68.66% | 67.81% |
| (GW, QW, LS, PS) | **69.87%** | **69.87%** | 67.97% | 66.09% |
| (GW, QW, LS, PW) | 63.08% | 63.08% | **63.90%** | 62.84% |
| (GS, QW, LW, PS) | **72.78%** | **72.78%** | 70.17% | 70.23% |
| (GS, QW, LW, PW) | **69.17%** | **69.17%** | 67.30% | 66.59% |
| (GW, QW, LW, PS) | **68.33%** | **68.33%** | 66.45% | 64.75% |
| (GW, QW, LW, PW) | **63.32%** | **63.32%** | 62.41% | 61.07% |

We next report the results on the MMLU dataset in Table 6. On this dataset, we find that OW-L, OW-I, and ISP outperform MV across all ensembles.

Table 6: Experimental Results: We report the overall accuracy for different LLM ensembles on MMLU. In the tables, the best results are highlighted in **bold**.

| Accuracy | OW-L | OW-I | ISP | MV |
|---|---|---|---|---|
| (GS, QS, LS, PS) | **90.37%** | **90.37%** | 90.01% | 89.32% |
| (GS, QS, LS, PW) | 89.59% | **89.74%** | 89.26% | 87.09% |
| (GW, QS, LS, PS) | **87.62%** | 87.57% | 87.08% | 86.06% |
| (GW, QS, LS, PW) | **85.36%** | **85.36%** | 83.33% | 82.60% |
| (GS, QS, LW, PS) | 90.42% | **90.51%** | 90.20% | 89.57% |
| (GS, QS, LW, PW) | **89.63%** | 89.61% | 88.99% | 86.82% |
| (GW, QS, LW, PS) | 87.90% | **87.91%** | 87.23% | 86.01% |
| (GW, QS, LW, PW) | 84.72% | **84.75%** | 82.88% | 81.38% |
| (GS, QW, LS, PS) | **89.03%** | **89.03%** | 88.57% | 87.49% |
| (GS, QW, LS, PW) | **87.27%** | **87.27%** | 85.14% | 84.26% |
| (GW, QW, LS, PS) | **84.58%** | **84.58%** | 83.92% | 83.57% |
| (GW, QW, LS, PW) | **80.82%** | 80.77% | 80.45% | 79.98% |
| (GS, QW, LW, PS) | 89.35% | **89.38%** | 88.64% | 87.55% |
| (GS, QW, LW, PW) | 86.73% | **86.77%** | 84.75% | 83.26% |
| (GW, QW, LW, PS) | 84.49% | **84.80%** | 84.03% | 82.89% |
| (GW, QW, LW, PW) | 80.10% | **80.39%** | 79.40% | 78.18% |

Finally, we present the experimental results on the ARMMAN dataset. From Table 7, we observe that when selecting GPT-35-turbo-0125, Qwen2.5-Instruct-14B, Llama3.2-1B-Instruct and Phi-4-mini-instruct, OW-L underperforms MV. Similarly, when selecting GPT-35-turbo-0125, Qwen2.5-Instruct-3B, Llama3.2-1B-Instruct and Phi-4, OW-I underperforms MV. Moreover, MV outperforms both OW-I and ISP when selecting GPT-35-turbo-0125, Qwen2.5-Instruct-3B, Llama3.1-8B-Instruct and Phi-4-mini-instruct. In all remaining cases, however, our three algorithms achieve higher accuracy than MV.

Table 7: Experimental Results: We report the overall accuracy for different LLM ensembles on ARMMAN. In the tables, the best results are highlighted in **bold**.

| Accuracy | OW-L | OW-I | ISP | MV |
|---|---|---|---|---|
| (GS, QS, LS, PS) | **85.78%** | **85.78%** | **85.78%** | 85.24% |
| (GS, QS, LS, PW) | **85.38%** | 85.35% | 85.35% | 83.41% |
| (GW, QS, LS, PS) | 84.30% | **84.57%** | 82.77% | 78.96% |
| (GW, QS, LS, PW) | 77.92% | **81.04%** | 79.50% | 75.74% |
| (GS, QS, LW, PS) | **85.78%** | **85.78%** | **85.78%** | 84.83% |
| (GS, QS, LW, PW) | **85.35%** | **85.35%** | **85.35%** | 82.55% |
| (GW, QS, LW, PS) | 84.30% | **84.59%** | 82.32% | 78.76% |
| (GW, QS, LW, PW) | 74.48% | **77.46%** | 77.37% | 74.99% |
| (GS, QW, LS, PS) | 84.89% | 84.89% | **85.18%** | 81.33% |
| (GS, QW, LS, PW) | **85.32%** | 81.16% | 80.64% | 78.08% |
| (GW, QW, LS, PS) | **84.30%** | 73.36% | 73.31% | 72.72% |
| (GW, QW, LS, PW) | **74.48%** | 68.85% | 68.82% | 69.06% |
| (GS, QW, LW, PS) | **84.63%** | 84.58% | 83.66% | 78.00% |
| (GS, QW, LW, PW) | **85.32%** | 79.75% | 80.06% | 74.73% |
| (GW, QW, LW, PS) | **84.30%** | 68.53% | 72.16% | 70.10% |
| (GW, QW, LW, PW) | **74.48%** | **74.48%** | 67.06% | 65.35% |

## G OMITTED DETAILS IN APPENDIX C

### G.1 PROOF OF THEOREM 4

For any $a_1, \cdots, a_N, s$, We now have

$$\mathbb{P}(S^* = s \mid A_1 = a_1, \cdots, A_N = a_N) = \frac{\mathbb{P}(A_1 = a_1, \cdots, A_N = a_N \mid S^* = s)\mathbb{P}(S^* = s)}{\mathbb{P}(A_1 = a_1, \cdots, A_N = a_N)}$$

$$= \mathbb{E}_{\alpha \sim D(\alpha)} \frac{\prod_{i:a_i=s} e^{\alpha\beta_i}}{\prod_{i=1}^N (K - 1 + e^{\alpha\beta_i})} \times \frac{\mathbb{P}(S^* = s)}{\mathbb{P}(A_1 = a_1, \cdots, A_N = a_N)}$$

$$= \mathbb{E}_{\alpha \sim D(\alpha)} \frac{\prod_{i:a_i=s} e^{\alpha\beta_i}}{\prod_{i=1}^N (K - 1 + e^{\alpha\beta_i})} \times \frac{\mathbb{P}(S^* = s)}{\mathbb{P}(A_1 = a_1, \cdots, A_N = a_N)}$$

$$= \mathbb{E}_{\alpha \sim D(\alpha)} \frac{e^{\alpha \sum_{i:a_i=s} \beta_i}}{\prod_{i=1}^N (K - 1 + e^{\alpha\beta_i})} \times \frac{\mathbb{P}(S^* = s)}{\mathbb{P}(A_1 = a_1, \cdots, A_N = a_N)}.$$

Since the Bayesian optimal aggregator chooses $\widehat{s} = \arg\max_s \mathbb{P}(S^* = s | A_1 = a_1, ..., A_N = a_N)$, we then only need to compare among $\sum_{i:a_i=s} \beta_i$ for all $s$. It is equivalent to aggregate as

$$\widehat{s} = \arg\max_{s \in S} \sum_{i=1}^N \beta_i \mathbb{1}\{a_i = s\},$$

which concludes the proof.

### G.2 PROOF OF THEOREM 5

We now show that, with the existence of $\alpha$, it still holds that ISP outperforms MV, which in turn outperforms SP. Note that we assume we don't know the difficulty level $\alpha$ for each question; otherwise, we can always group questions with the same $\alpha$. We study the case where there are only two labels. We use $x_i(\alpha)$ to denote the accuracy under difficulty $\alpha$.

First, we have the following equations,

$$\mathbb{P}(A_1 = s_i | A_2 = s_j) = \mathbb{E}_{D(\alpha)} \frac{x_1(\alpha)(1 - x_2(\alpha)) + (1 - x_1(\alpha))x_2(\alpha)}{K - 1} + \frac{K - 2}{(K - 1)^2}(1 - x_1(\alpha)(1 - x_2(\alpha))),$$

for any $s_i \neq s_j$ and

$$\mathbb{P}(A_1 = s_i | A_2 = s_i) = \mathbb{E}_{D(\alpha)} x_1(\alpha)x_2(\alpha) + \frac{(1 - x_1(\alpha))(1 - x_2(\alpha))}{K - 1}.$$

Recall that for any question, we use $s^*$ to denote the true label. For this question, due to the linearity of the expectation, we only need to prove that for any $\alpha$ and $\alpha'$, it holds that

$$g(\beta_1, \beta_2) = x_2(\alpha)[\frac{x_1(\alpha')(1 - x_2(\alpha')) + (1 - x_1(\alpha'))x_2(\alpha')}{K - 1} + \frac{K - 2}{(K - 1)^2}(1 - x_1(\alpha')(1 - x_2(\alpha')))]$$

$$+ \frac{1 - x_2(\alpha)}{K - 1}[x_1(\alpha')x_2(\alpha') + \frac{(1 - x_1(\alpha'))(1 - x_2(\alpha'))}{K - 1}]$$

$$+ (1 - x_2(\alpha))\frac{K - 2}{K - 1}[\frac{x_1(\alpha')(1 - x_2(\alpha')) + (1 - x_1(\alpha'))x_2(\alpha')}{K - 1} + \frac{K - 2}{(K - 1)^2}(1 - x_1(\alpha')(1 - x_2(\alpha')))]$$

$$- \frac{1}{K} \le 0.$$

Recall that $x_i = \frac{e^{\alpha\beta_i}}{e^{\alpha\beta_i} + K - 1}$ for any $\alpha$. Therefore, it holds that

$$g(\beta_1, \beta_2) = \frac{e^{\alpha'\beta_1} + e^{\alpha'\beta_2} + K - 2}{(K - 1 + e^{\alpha'\beta_1})(K - 1 + e^{\alpha'\beta_2})}(\frac{e^{\alpha\beta_2}}{(K - 1)(K - 1 + e^{\alpha\beta_2})} + \frac{K - 2}{K - 1})$$

$$+ \frac{1}{(K - 1 + e^{\alpha'\beta_2})(K - 1 + e^{\alpha\beta_2})}(\frac{e^{\alpha'(\beta_1 + \beta_2)} + K - 1}{K - 1 + e^{\alpha'\beta_1}}) - \frac{1}{K}.$$

Notice that when $\beta_1 = 0$, we have $g(0, \beta_2) = 0$. We now calculate the partial derivative of $g$ with respect to $\beta_1$. It holds that

$$\frac{\partial g}{\partial \beta_1} = \frac{\alpha' e^{\alpha'\beta_1}(e^{\alpha\beta_2} - 1)(1 - e^{\alpha'\beta_2})}{(K - 1 + e^{\alpha\beta_2})(K - 1 + e^{\alpha'\beta_2})(K - 1 + e^{\alpha'\beta_1})^2} \le 0.$$

Therefore, we know that $g(\beta_1, \beta_2) \le g(0, \beta_2) = 0$ as $\beta_1 \ge 0$ holding for any difficulties $\alpha$ and $\alpha'$. By symmetry, it suffices to average over the other agents, which does not alter the fact that the expected advantage of ISP remains greater than that of MV. We hence conclude that $\mathbb{E}[Adv_{ISP}(s^*)] \ge \mathbb{E}[Adv_{MV}(s^*)]$ similar as the proof of Theorem 2.

Similarly, when we compare MV with SP, we only need to prove

$$h(\beta_1, \beta_2) = x_2(\alpha)[x_1(\alpha')x_2(\alpha') + \frac{(1 - x_1(\alpha'))(1 - x_2(\alpha'))}{K - 1}]$$

$$+ (1 - x_2(\alpha))[\frac{x_1(\alpha')(1 - x_2(\alpha')) + (1 - x_1(\alpha'))x_2(\alpha')}{K - 1}$$

$$+ \frac{K - 2}{(K - 1)^2}(1 - x_1(\alpha')(1 - x_2(\alpha')))] - \frac{1}{K} \ge 0.$$

It then holds that

$$h(\beta_1, \beta_2) = \frac{e^{\alpha\beta_2}}{K - 1 + e^{\alpha\beta_2}}\frac{e^{\alpha'(\beta_1 + \beta_2)} + K - 1}{(K - 1 + e^{\alpha'\beta_1})(K - 1 + e^{\alpha'\beta_2})}$$

$$+ \frac{K - 1}{K - 1 + e^{\alpha\beta_2}}\frac{e^{\alpha'\beta_1} + e^{\alpha'\beta_2} + K - 2}{(K - 1 + e^{\alpha'\beta_1})(K - 1 + e^{\alpha'\beta_2})} - \frac{1}{K}.$$

Notice that when $\beta_1 = 0$, it holds that $h(0, \beta_2) = 0$ for all $\beta_2$. In addition, with some calculations, it holds that

$$\frac{\partial h}{\partial \beta_1} = \frac{\alpha' e^{\alpha'\beta_1}(K - 1)(e^{\alpha\beta_2} - 1)(e^{\alpha'\beta_2} - 1)}{(K - 1 + e^{\alpha\beta_2})(K - 1 + e^{\alpha'\beta_2})(K - 1 + e^{\alpha'\beta_1})^2} \ge 0.$$

Therefore, we know that $h(\beta_1, \beta_2) \ge h(0, \beta_2) = 0$ for all $\beta_2 \ge 0$, yielding $\mathbb{E}[Adv_{SP}(s^*)] \le \mathbb{E}[Adv_{MV}(s^*)]$ due to the linearity of expectation.

Despite the difficulty of questions being unknown in general LLM reasoning settings, ISP still enjoys a larger expected advantage compared to majority voting. This demonstrates the robustness of our approach in more general scenarios and highlights its broad applicability.

## H    THE USE OF LARGE LANGUAGE MODELS (LLMS)

We only use LLMs for simple writing checks, such as grammar errors. Since LLMs do not play a significant role, they should not be regarded as a contributor. This concludes the appendix.

