# OpenReview forum: "Beyond Majority Voting: LLM Aggregation by Leveraging Higher-Order Information"
_ICLR.cc/2026/Conference — Submitted to ICLR 2026_

### Official Review · Reviewer_1dqM · 2025-10-19

**Soundness:** 3
**Presentation:** 1
**Contribution:** 2
**Rating:** 4
**Confidence:** 3

**Summary:**

This paper proposes two methods—Optimal Weight (OW) and Inverse Surprisingly Popular (ISP)—to improve aggregation of answers from multiple LLMs beyond simple majority voting. OW leverages first-order information (model accuracies) and is provably Bayesian-optimal when accuracies are known. ISP uses second-order information (prediction correlations) without requiring ground-truth labels and outperforms both majority voting and the classic Surprisingly Popular method.

I have some concerns about the paper at this stage, so my evaluation may be biased towards the negative. If the author can address these issues in the rebuttal stage, I will consider raising my score.

**Strengths:**

1. The concepts proposed are novel.

2. Achieved good performance on multiple datasets

**Weaknesses:**

1. Related work is an important part of the paper and should not be put in the appendix.

2.  Why is the LLM's output unaffected by the order of the options? Is there any literature or experiment demonstrating this? Since different letters have different statistical probabilities, the LLM should have different preferences for different letters.

3. Why didn't the authors conduct experiments on widely used mathematical and common sense reasoning datasets, such as GSM 8K? This might be more helpful in verifying the effectiveness of the proposed method.

4. I'm sorry that I can't find any relevant literature, but in my impression, optimal weight aggregation should not be the first one proposed in this paper. It has been proposed in other works before.

**Questions:**

1. Are the concepts of first-order information and second-order information first defined in this article?

2. How to obtain the expected accuracy？

3. The sizes of the several models selected in the experiment are obviously unbalanced. GPT-4o is much larger than the other models. Will this affect the experimental results?

---

> ### Author Response · Authors · 2025-11-19
>
> Thank you for your kind remarks and questions. We believe our response below should help thoroughly address the reviewer's major questions in order, and are happy to engage with the reviewer if there is any further question.
>
> Re W1 **Related Work**: We thank the reviewer for the suggestion. To address this concern, we have already added back the related work section to the main text from the appendix.
>
> Re W2 **The order of the options**: This claim is supported by several pieces of evidence. First, we randomly shuffle the order of the options, so each option is equally likely to contain the correct answer in expectation.
> Second, prior work has shown that modern LLMs exhibit strong long-context capabilities, and their predictions are largely invariant to superficial changes such as option order (see Lines 149–154 and [1] cited therein).
> Finally, our experiments confirm that LLMs do not display noticeable positional or letter-based biases. For example, on the ARMMAN Healthcare dataset—which the models have never seen—GPT-4o selects option A 49.7% of the time and option B 50.3%. Among open-source models, Qwen2.5-Instruct-14B chooses A 51.4% and B 48.6%. These nearly balanced distributions provide empirical evidence that option order does not materially influence model outputs.
>
> [1] Xiaobo Guo and Soroush Vosoughi. Serial position effects of large language models. arXiv preprint arXiv:2406.15981, 2024.
>
> Re W3 **The selection of datasets**: Our theoretical results are derived for the finite $K$-option multiple-choice setting, whereas GSM8K consists of free-form math word problems. For this reason, we selected MMLU—which contains a substantial number of mathematical questions—as a more appropriate benchmark within the $K$-option framework considered in the paper. Extending our methods to open-response reasoning tasks such as GSM8K is an interesting and promising direction for future work.
>
> Re W4 **Novelty of the optimal weight aggregation**: The reviewer is correct that the idea of optimal weight aggregation was not first proposed in our paper, and we actually provided an extensive literature review about information aggregation in Appendix A.1.
> However, our contribution and novelty lie in being the first to formulate a natural structure for the LLM inference setting in leverage of specific LLM characteristics, which then allows us to develop closed-form optimal weights. This is impossible in general aggregation tasks.  Prior concurrent LLM studies [1,2,3] predominantly rely on simple majority voting rather than principled weight estimation. Hence we believe that bringing information-theoretic tools into multi-agent LLM reasoning opens a novel and meaningful direction for improving LLM performance.
>
> [1] Jiyi Li. A comparative study on annotation quality of crowdsourcing and llm via label aggregation. In ICASSP 2024-2024 IEEE International Conference on Acoustics, Speech and Signal Processing (ICASSP), pp. 6525–6529. IEEE, 2024.
>
> [2] Eray Can Elumar, Cem Tekin, and Osman Yagan. Cost-aware llm-based online dataset annotation. arXiv preprint arXiv:2505.15101, 2025.
>
> [3] Vighnesh Subramaniam, Yilun Du, Joshua B Tenenbaum, Antonio Torralba, Shuang Li, and Igor Mordatch. Multiagent finetuning: Self improvement with diverse reasoning chains. arXiv preprint
> arXiv:2501.05707, 2025.

---

> ### Author Response · Authors · 2025-11-19
>
> Re Q1 **Higher-order information**: These definitions were not first proposed by us, and we cited relevant prior work (lines 241-242).
> More importantly, our work is the first to **connect first-order and second-order information concepts to the characteristics of LLMs** and to demonstrate that leveraging higher-order information can effectively improve LLM inference accuracy. Our analysis also reveals that the classical second-order method SP performs worse than MV for LLMs (see Lines 285–296), which motivates our ISP method that adapts SP to the LLM inference regime. This integration of information-theoretic structure with multi-agent LLM reasoning is, to our knowledge, novel.
>
> Re Q2 **Obtaining the accuracy**: We provide two ways to estimate the expected accuracies using second-order information.
>
>  (1) OW-L: In Eq. (7), we use an ERM procedure to learn $x_i$ by minimizing the $L_2$ loss between the empirical conditional probabilities and the model-implied conditional probabilities (which depend on $x_i$).
>
>  (2) OW-I: As described in Lines 443–449, we can use ISP to generate an aggregated pseudo-label and treat it as the ground truth for estimating accuracies.
>
> Both OW-L and OW-I rely on these second-order–based estimators and achieve strong empirical performance, demonstrating that accurate expected accuracy estimation is feasible in an unsupervised setting.
>
> Re Q3 **Unbalanced model sizes**: We do not expect model size differences to affect our conclusions.
> Although GPT-4o is larger, several of the other models (e.g., Qwen2.5-Instruct-14B) are more recent and have undergone strong post-training, resulting in **comparable capabilities**. For example, across the three datasets we evaluate, GPT-4o achieves accuracies of 73%, 91%, and 85%, while Qwen2.5-Instruct-14B achieves 72%, 89%, and 85%. Thus, in terms of **model ability**, the models are well balanced despite differences in parameter counts. Additional comparisons can be found in Figure 2.
> Moreover, our methods remain robust even when combining models with unbalanced abilities. We provide results for 16 combinations of strong and weak models in Tables 5, 6 and 7 (Appendix F.4).

---

> > ### Comment · Reviewer_1dqM · 2025-11-27
> > **Reply to the rebuttal**
> >
> > Thanks for the author's reply.
> >
> > I have increased the presentation score to 3. Additionally, I noticed Reviewer 96ii's comments, which I think are very insightful. I will follow your discussions and reserve the possibility of changing the score.

---

> > > ### Author Response · Authors · 2025-11-27
> > >
> > > Thank you for your updated feedback and for increasing the presentation score. We have clarified the misunderstandings raised by Reviewer 96ii and added further experiments to support our statements. We believe all concerns have now been fully addressed and hope you will consider adjusting the overall score accordingly.

---

### Official Review · Reviewer_2A2J · 2025-10-30

**Soundness:** 3
**Presentation:** 3
**Contribution:** 2
**Rating:** 4
**Confidence:** 3

**Summary:**

This paper investigates how to aggregate answers from multiple Large Language Model (LLM) agents to overcome the limitations of standard Majority Voting (MV). The authors note that MV ignores heterogeneity and correlation among models. To address this, the paper proposes two new algorithms that leverage higher-order information, namely Optimal Weight (OW) and Inverse Surprising Popularity (ISP). Since OW only need unknown true accuracies, the authors further propose two practical unsupervised methods (OW-L and OW-I). The experimental section validates these methods on synthetic datasets, UltraFeedback, MMLU and ARMMAN.

**Strengths:**

How to effectively aggregate the response from several LLMs is important tasks. This paper goes beyond simple heuristics, formally modeling the problem from an information-theoretic perspective is interesting.

**Weaknesses:**

1. The paper describes MoE as "only one expert gets a non-zero weight", which is too simplification. Modern MoE combines outputs from multiple experts via weighted averaging. The authors should more accurately describe the distinction from their work (e.g., aggregating final answers vs. internal representations).
2. The authors should explain why in Table 3, the 'Single Best' model on the MMLU dataset outperforms the proposed aggregation methods.
3. The proposed method optimizes the aggregator on several large datasets. But does it generalize effectively to other, unseen datasets?
Questions: Please refer to weakness.

**Questions:**

Theorem 3 shows that the dataset size (M) needs to be large to reduce the error. However, considering that N LLMs are required, is the cost of this algorithm still prohibitively high?

---

> ### Author Response · Authors · 2025-11-19
>
> Thank you for your kind remarks and questions. We believe our response below should help thoroughly address the reviewer's major questions in order, and are happy to engage with the reviewer if there is any further question.
>
> Re W1 **MOE discription**: We thank the reviewer for the suggestion. We agree that modern MoE architectures may activate multiple experts (e.g., top-$k$) and average their internal representations through a gating mechanism. This process is fundamentally different from our setting, where we aggregate **final answers** rather than **internal representations**. Moreover, MoE gating does not leverage higher-order information such as inter-expert correlations, whereas our method explicitly incorporates such information from an information-theoretic perspective. We have revised the description in the revised version to more accurately reflect these distinctions.
>
> Re W2 **Comparing single best and our methods**: We would like to emphasize that the “Single Best’’ result in Table 3 is not a strictly comparable baseline (should be viewed as a clairvoyant oracle) because it relies on **additional information**—namely, knowing which model achieves the highest accuracy on the dataset, so it does not indicate that our method fails on the MMLU dataset (See discussions in Lines 481-484). This requires labeled data, whereas our methods operate in a **fully unsupervised** setting with no access to labels. Under such conditions, the identity of the best model is unknown and cannot be directly selected. Moreover, even when labels are available (and thus the Single Best model is artificially revealed), our method still outperforms or matches it on other datasets such as UltraFeedback and ARMMAN, demonstrating the robustness and practical benefit of our approach.
>
> Re W3 **Generalization of our methods**: Our evaluations are conducted in a fully **unsupervised**, **training-free**, **zero-shot** setting: the LLMs never observe any labeled answers from the datasets used in our experiments. In this sense, every test is performed on an unseen dataset.
> If the concern is that the LLMs might have encountered some public benchmarks during pretraining, we address this by evaluating on the **ARMMAN Healthcare dataset** (lines 470–477), which is a recent non-public dataset that the models are unlikely to have seen during pretraining. On this dataset as well, our methods outperform both MV and Single Best, demonstrating strong generalization beyond the standard benchmarks.
>
> Re Q1 **The cost of our algorithm**: We don’t think this will be a major issue. First of all, the $1/\sqrt{M}$ estimation error in Theorem 3 is a fundamental statistical limit—any empirical estimation, even of a simple mean, necessarily exhibits this rate.
> More importantly, in practice, the convergence is quite fast. Moreover, LLM inference is efficient: on a single A100 GPU, generating over 60,000 UltraFeedback responses with a 14B model takes roughly 30 minutes, and smaller models run even faster. This corresponds to an average cost of about 0.1 seconds per query when using $N=4$ models, which is entirely practical. This cost is negligible compared to training-based aggregation methods, which may require multiple GPUs running for days.
> Also, our aggregation algorithm itself requires **no training**. If inference is performed via an API, the entire workflow— including OW-I/OW-L—can be executed on a CPU without any GPU involvement. OW-L and OW-I's ability-estimation step takes a few seconds on the UltraFeedback dataset (OW-I: 6.1s, OW-L: 10.3s), while the second-stage aggregation takes less than 0.1s. All experiments were run locally on a MacBook Pro (Apple M3 Pro). Since the dominant cost in modern LLM pipelines typically comes from gradient-based training, our training-free approach avoids this bottleneck entirely. Enhancing the reasoning and aggregation stage in this way allows individuals and small organizations, who may lack large compute resources but can easily perform inference, to benefit from LLM improvements and contribute more effectively to the community.

---

### Official Review · Reviewer_TNoU · 2025-10-30

**Soundness:** 2
**Presentation:** 3
**Contribution:** 2
**Rating:** 4
**Confidence:** 3

**Summary:**

This paper studies the response aggregation problem of multi-agent large language models (LLMs), arguing that simple majority voting ignores the heterogeneity and correlation between models. The authors propose two new algorithms: (1) Optimal Weight, which uses first-order information, i.e., the accuracy of each LLM, for Bayesian optimal weighted aggregation; and (2) Inverse Surprising Popularity, which uses second-order information, i.e., the correlation between the answers provided by each agent, to improve the traditional surprising popularity method. Theoretical analysis proves the Bayesian optimality of OW and the advantage of ISP over MV. Experiments validate the methods on synthetic data, UltraFeedback, MMLU, and ARMMAN datasets, showing an improvement of 0.5-2 percentage points compared to MV. However, the problem studied in this paper suffers from insufficient importance, and the practical value of the methods is quite limited.

**Strengths:**

S1. The core theoretical results of the paper, including the Bayesian optimality of OW (Theorem 1) and the expected advantage of ISP over MV (Theorem 2), are mathematically rigorous.

S2. The authors found that the traditional surprising popularity method performed worse than simple MV in LLM scenarios. The authors' explanation makes sense: SP is designed to correct systematic biases in human judgment, but LLM has less systematic bias, so SP can only take advantage of a smaller margin of error.

S3. The experimental design of this paper is comprehensive, including three levels: (1) synthetic data that fully conforms to the theoretical assumptions to verify the theoretical predictions; (2) standard LLM benchmarks to verify practicality; and (3) real-world medical and health applications (ARMMAN) to demonstrate potential social value. The authors also used eight models from four families including GPT, Qwen, Llama, and Phi to form 16 combinations that cover models of different scales and capabilities.

S4. Appendix B.2’s relaxation of the conditional independence assumption is a positive attempt.

**Weaknesses:**

W1. The fundamental problem is that the research question itself is not important enough, and its practical value is limited. Specifically, the paper devotes a large portion to studying how to optimally aggregate votes from multiple LLMs, but this problem has very limited importance in the current LLM application ecosystem.

First, the closed-set classification (K fixed options) that the paper focuses on is only a small part of LLM applications; the more mainstream applications are open-ended generation, multi-turn dialogue, and complex reasoning, and the paper's method cannot be extended to these scenarios.

Second, the improvement shown in experiments is extremely small (0.5-2 percentage points), not worth increasing the system complexity.

Third, in practice, there are often simpler and more effective solutions like directly use the strongest single model (Table 3 shows that a single model is sometimes better than ensemble), or invest in training a better model, rather than ensemble multiple weak models.

Fourth, the core issue in multi-agent LLM reasoning is how to design an effective debate protocol and how to enable models to truly exchange and integrate information and the final voting is actually the most trivial part. The paper is essentially optimizing a marginal problem.

W2. The OW algorithm is a beautiful theoretical contribution, proving its Bayesian optimality. However, this method relies on knowing the accuracy of each LLM, and in the unsupervised aggregation scenario considered in the paper, obtaining these accuracies requires a large amount of labeled data. Since it is unsupervised, how do we obtain this data?

W3. Theorem 2 shows that the advantage of ISP decays at a rate of 1/K, meaning that for tasks with many options or open-ended functions, ISP offers virtually no advantage. The paper acknowledges this limitation in its conclusion but does not propose a solution. This may further restricts the applicability of the method.

W4. The paper does not adequately discuss the limitations and applicability of the method.

**Questions:**

Q1. The paper focuses on the final aggregation step, but is this really the most important bottleneck in multi-agent LLM reasoning?

Q2. Table 3 shows that the single strongest model achieves 91.02% on MMLU, while the ensemble only achieves 90.37%. This seems to suggest that the value of ensemble is questionable. Could you provide a cost-benefit analysis comparing `ensemble of 4 intermediate models` vs. `using only one strongest model` vs. `using the budget of ensemble to train/fine-tune a better single model`?

Q3. How can these methods be extended to open-ended tasks?

Q4. What are the theoretical guarantees of the OW-L and OW-I methods?

Q5. ISP requires estimating O(N²K²) conditional probabilities, which becomes a bottleneck when N and K are large. How can this be scaled?

---

> ### Author Response · Authors · 2025-11-19
>
> Thank you for your kind remarks and questions. We believe our response below should help thoroughly address the reviewer's major questions in order, and are happy to engage with the reviewer if there is any further question.
>
> Re W1:
>
> **K-fixed options**: We respectfully disagree that the problem with k-fixed options "is not important".
> First of all, many important LLM applications do operate in fixed-option settings. For instance, in our Healthcare dataset, the task is to predict  pregnant and postpartum women are at risk of dropping out from call-based maternal health information (a binary outcome), for which we do believe any additional accuracy improvement is very valuable.
> Another important example -- also a core motivation for our work -- is the automatic construction of preference datasets for fine-tuning algorithms like DPO. Here, principled aggregation of preferences remains essential to today's fine-tuning algorithms.
> A third example is that numerous forecasting tasks (e.g., whether the Fed will decrease interest rate, or not) are multiple-options as well, and improving accuracy of forecasting is well-known to be an important problem.
> While open-ended generation is indeed a key research direction, we believe **allowing the pursuit of diverse ideas and research routes is crucial for continuous innovations in AI** and its ultra success. Our study of LLM aggregation in these important applications is novel, and we believe that such foundational progress opens new avenues for future research.
>
> **Experimental improvement**: We respectfully disagree that the reported improvements are insignificant. Our method delivers measurable and **training-free** accuracy gains entirely **without any additional LLM training or labeled data**, making it both efficient and practical.
> In contrast, post-training or supervised alignment techniques require substantial GPU resources; our approach, by comparison, only involves lightweight reasoning and higher-order information computation that can be performed on a CPU.
> Also, the seemingly small improvements (0.5–2 percentage points) should be interpreted in the context of strong base models. For example, on MMLU, the LLM already achieves around 90% accuracy; a 1% improvement corresponds to resolving roughly **10% of the remaining errors**, which is meaningful. Moreover, when evaluated on weaker LLMs, the performance gains become substantially larger. As shown in Lines 517–519, our method can improve accuracy by up to **14.20%**, which is remarkable for a fully training-free approach.
>
> **Why ensembling**: We would like to emphasize that “just use the strongest model” is neither reliable nor practical in real-world scenarios. At the practical level, we often **do not know which model is the strongest** -- and this very often depends on the problem at hand (see Lines 481-484). For example, GPT-35-turbo substantially outperforms Llama3.1-8B-Instruct on MMLU (77% vs. 72%), performs significantly worse on our Healthcare dataset (54% vs. 62%), and yields nearly identical results on UltraFeedback (59% vs. 59%).
> Our OW-I and OW-L methods directly address it: their first stage uses second-order information to estimate each model’s underlying ability, providing a principled way (utilizing higher-order information) to identify strong models. This contributes not only to aggregation but also to the broader problem of model selection.
> More importantly, compared with investing in training a better model, our approach is **completely training-free**, offering substantial computational advantages. Beyond the generation step—which requires no GPU when using APIs—the aggregation itself requires no GPU resources and can be run entirely on a CPU or personal computer. This makes the method practical for settings with limited compute, enabling small organizations and individual users to benefit from LLM ensemble gains without expensive training infrastructure.

---

> ### Author Response · Authors · 2025-11-19
>
> **Importance of voting**: We respectfully disagree with the view that the aggregation step is trivial. Enabling models to exchange information does not diminish the importance of how their outputs are ultimately integrated. Aggregation is the key mechanism through which diverse reasoning paths are combined into a final outcome. Neglecting this step risks losing valuable information produced by the agents. Recent work (e.g., [1]) has similarly emphasized that principled weighting and aggregation are essential for reliable multi-agent reasoning. Our contribution advances this direction by introducing an information-theoretic higher-order weighting mechanism that provides both theoretical guarantees and deeper interpretability.
> Moreover, the focus on aggregation is complementary—not contradictory—to research on debate protocols. In multi-round debate systems, prior work such as [2] primarily studies post-training, whereas our method operates purely at the reasoning level. Integrating our improved aggregation module into multi-turn debate protocols represents a promising direction for future research, and our results suggest that enhancing the voting component can meaningfully strengthen such systems.
>
> [1] Yichao Fu, Xuewei Wang, Yuandong Tian, and Jiawei Zhao. Deep think with confidence. arXiv preprint arXiv:2508.15260, 2025.
>
> [2] Vighnesh Subramaniam, Yilun Du, Joshua B Tenenbaum, Antonio Torralba, Shuang Li, and Igor Mordatch. Multiagent finetuning: Self improvement with diverse reasoning chains. arXiv preprint arXiv:2501.05707, 2025.
>
> Re W2 **Learning accuracies with unlabelled data**: We would like to clarify that our framework is explicitly designed to overcome the challenge of estimating model accuracies without labeled data. This concern directly motivates our OW-I and OW-L algorithms.
> In the first stage, we estimate each model’s accuracy using **second-order information**, i.e., the correlation structure among model outputs. This procedure is entirely **unsupervised** and does not rely on any labeled data. The ability-estimation step is itself a meaningful contribution, as it provides an effective way to identify strong models without supervision. Concretely, for OW-L, we can perform ERM in Eq. (7) to learn $x_i$ by minimizing the $L_2$ loss between the empirical conditional probabilities and the model-implied conditional probabilities, both of which depend on $x_i$. Alternatively, for OW-I, as described in Lines 443–449, we can use ISP to produce an aggregated pseudo-label and treat it as the ground truth for estimating accuracies. Both OW-L and OW-I follow this unsupervised estimation strategy and achieve strong empirical performance, demonstrating that accurate ability estimation is feasible without labeled data.
>
> Re W3 **Solutions when $K$ is large**: We clarify that while the theoretical advantage of ISP scales as $1/K$, this effect is minor in realistic LLM applications. In practice, the effective number of options $K$ is typically small because modern LLMs exhibit low perplexity and their outputs cluster into a limited number of coherent categories. Under these conditions, ISP remains a reliable and accurate estimator of model accuracies.
> Moreover, our framework provides a principled path forward even when $K$ increases. ISP (or other related estimators) can first be used for ability estimation, after which the OW procedure computes optimal aggregation weights. We note that the optimality of the OW algorithm itself is **independent of $K$**: once the accuracies $x_i$ are estimated, OW yields the optimal weights regardless of the number of options. This modular design retains the theoretical optimality of OW while mitigating the $1/K$ decay inherent to ISP alone.
>
> Re W4 **Limitations and applicablity**: We thank the reviewer for this comment. We discuss the main limitations of our approach in the conclusion section, and now expanded this discussion to more clearly articulate the applicability and potential extensions of our method. These clarifications will be included in the camera-ready version.
>
> Re Q1: **Why aggregation is important**: Please refer to our reply to W1 for a discussion on the importance of aggregation. We would like to emphasize again that the key advantage of our approach is that it offers **performance gains at extremely low cost**. In an era where pretraining and post-training are increasingly constrained by computational bottlenecks and consume significant resources, our aggregation method is essentially *free*: it requires no labeled data, no gradient computation, and can be executed efficiently on a single CPU. This makes the technique broadly accessible and enables many smaller groups and individual practitioners—who cannot afford large-scale training—to benefit from advances in multi-agent LLM reasoning. We believe this low-cost, training-free improvement provides practical value to the community and complements more resource-intensive directions such as LLM debate.

---

> ### Author Response · Authors · 2025-11-19
>
> Re Q2 **Our aggregation methods are extremely low-cost**: Please refer to our response to W1. Identifying the strongest single model is itself a nontrivial and data-dependent problem. It requires an estimation procedure, and our OW-I/OW-L first stage—based on higher-order information—is designed precisely for this purpose. Even then, this estimation can occasionally be imperfect due to data limitations. For example, on the MMLU dataset, our analysis shows that in nearly **25% of model combinations we fail to identify the true strongest model**. This demonstrates that comparing against the “single best model” is largely a theoretical construct but **difficult to achieve in practice**.
> Because it uses additional information, we have clarified in the revised version that the “Single Best” should be viewed as a **clairvoyant oracle** rather than a fair baseline (see Lines 481–484).
> Also, regarding the cost comparison, the computational cost of aggregation is almost negligible. For example, on the UltraFeedback dataset, OW-L and OW-I's ability-estimation step takes a few seconds(OW-I: 6.1s, OW-L: 10.3s), while the second-stage aggregation takes less than 0.1s. All experiments were run locally on a MacBook Pro (Apple M3 Pro). The results are similar for other datasets, indicating that both algorithms can be easily executed entirely on a personal CPU. In contrast, training or fine-tuning a stronger single model is typically infeasible in practical settings: closed-source models such as GPT cannot be fine-tuned by users, and even for open-source models, fine-tuning demands substantial GPU resources—often hours or days on expensive hardware. Thus, the **cost of our approach is orders of magnitude lower**, while still offering measurable performance gains, making it a highly cost-effective alternative to model training.
>
> Re Q3 **Open-ended tasks**: Our approach can be extended to open-ended tasks by appropriately adapting the accuracy-estimation step. In particular, second-order information can still be leveraged by grouping model outputs into semantically coherent buckets—for example, clustering open-ended responses into $K$ categories (where, due to the increasingly low perplexity of modern LLMs, $K$ is typically not large), or we can use light human annotation. Once these accuracies are estimated, the OW framework can be applied to calculate optimal weights for different LLM agents.
>
> Re Q4 **Guarantee of OW-L and OW-I**: Our theoretical results imply both OW-L and OW-I enjoy asymptotic consistency guarantees. When the number of questions goes to infinity, the estimated accuracies converge to the true accuracies. More concretely, Theorem 3 shows that the estimation error of the conditional probabilities satisfies $| \hat{P} - P |\lesssim 1/\sqrt{M}$, where $M$ is the number of samples. As a result, the empirical objective in Eq. (7) converges to its population counterpart, and the optimizer  $\hat x$ converges to the true accuracy vector $x$. This provides a theoretical guarantee that OW-L and OW-I recover the correct accuracies in the large-sample limit.
>
> Re Q5 **No bottleneck for ISP**: We respectfully disagree this can be considered as a bottleneck in practice. Although the theoretical complexity of ISP is $O(N^2 K^2)$, in practice it is very efficient. The computation only involves simple counting and normalization operations, can be fully run on a CPU, and has very low memory requirements. In our experiments, we can process tens of thousands of examples within a few seconds on a standard machine (Apple M3 Pro), taking around 6 seconds for the Ultrafeedback dataset and a similar time for the MMLU and ARMMAN datasets. If needed, the procedure can also be trivially parallelized or batched, making it scalable to even larger settings.

---

### Official Review · Reviewer_96ii · 2025-11-01

**Soundness:** 3
**Presentation:** 2
**Contribution:** 2
**Rating:** 4
**Confidence:** 4

**Summary:**

This study addresses the fundamental challenge of **aggregating responses from multiple Large Language Models (LLMs)** to produce a single, reliable collective decision, aiming to significantly improve upon simple **majority voting (MV)** or reliance on a single LLM.


The paper proposes two key aggregation schemes:
- The **Optimal Weight (OW)** algorithm generates the most accurate collective decision by assigning a **vote weight ($\omega_i$)** to each LLM $i$ based on its **known expected accuracy** ($x_i$).
- A practical limitation of OW is that the true expected accuracies ($x_i$) are often **unavailable in real-world scenarios**. To overcome this, the study introduces the **Inverse Surprising Popularity (ISP)** algorithm. ISP bypasses the need for ground-truth labels by leveraging **predictions correlations** among LLMs to infer the correct prediction. ISP functions as an aggregation method in its own right and, critically, provides a mechanism to **estimate the optimal weights needed for OW** in label-free settings.


The algorithms were validated across simulated datasets, LLM benchmarks (UltraFeedback, MMLU), and a real-world healthcare dataset (ARMMAN). Both **OW and ISP consistently outperformed MV**. Ultimately, the best results were achieved by a heuristic combination of both OW and ISP approaches, which demonstrated **strictly higher accuracy than the single best individual LLM** on several real-world tasks.

**Strengths:**

- The design of the ISP aggregation scheme proposed by the study offers a practical, cost-effective solution to multi-agent LLM aggregation schemes because it relies only on correlations between LLM predictions. This capability makes ISP and related heuristics (OW-L, OW-I) directly applicable to unsupervised, real-world scenarios like automated data annotation.

**Weaknesses:**

- One of the assumptions of this study relies on the idea that we can model the difficulty of a prompt question given to an LLM using **α** (question difficulty) and **βᵢ** (model ability). However, these quantities are inherently **subjective** and cannot be reliably measured. This raises concerns about the realism and validity of the assumptions on which this work is based.

- The study’s **comparative evaluation** is also limited, as it primarily benchmarks the proposed algorithms against **MV**, a relatively simple baseline. More advanced multi-agent reasoning strategies such as the ones described in the related work section (see Appendix A.1) were not included in the comparison, constraining the empirical scope of the findings.

- Additionally, the paper **does not discuss statistical significance** or provide detailed information about its experimental setup. Key aspects such as, variance, and number of seeds used are not reported, making it difficult to assess the robustness and reproducibility of the results. Including these details would strengthen the credibility of the findings and provide clearer insight into the reliability of the proposed methods.

**Questions:**

- Why do OW‑L and OW‑I show nearly identical performance in the experiments (Tables 3–6)? Specifically, is there a correlation between the objective optimized in OW‑L and the metrics underpinning ISP, which OW‑I use to produce the weights?

---

> ### Author Response · Authors · 2025-11-19
>
> Thank you for your kind remarks and questions. We believe our response below should help thoroughly address the reviewer's major questions in order, and are happy to engage with the reviewer if there is any further question.
>
> Re W1 **Assumptions on difficulty and ability**:
> This is likely due to a major misunderstanding about our results. Note that in our main model and main results presented in the main text, we do **not** make any assumptions regarding question difficulties or model abilities. In the appendix, we generalized our framework to capture the situation where additional information about question difficulties and model abilities is available. This is the only place this extra modeling setup shows up, and it is standard in the literature (e.g., [1, 2]).
>
> [1] Jacob Whitehill, Ting-fan Wu, Jacob Bergsma, Javier Movellan, and Paul Ruvolo. Whose vote should count more: Optimal integration of labels from labelers of unknown expertise. Advances in neural information processing systems, 22, 2009.
>
> [2] Jiyi Li. A comparative study on annotation quality of crowdsourcing and llm via label aggregation. In ICASSP 2024-2024 IEEE International Conference on Acoustics, Speech and Signal Processing (ICASSP), pp. 6525–6529. IEEE, 2024.
>
> Re W2 **Comparative baselines**: We respectfully emphasize that MV is an appropriate and fair benchmark for evaluating our methods.
> Prior work ([2,3,4]) typically applies majority voting for aggregation and then focuses on post-training improvements. In contrast, our work conducts an in-depth study of the *aggregation step itself* in multi-agent settings, showing that the aggregation component—often treated as a simple preprocessing step in previous studies—actually has substantial untapped potential for improvement.
> At the same time, our setting adheres to a fully **unsupervised** paradigm: models output answers without access to any external reward model or labeled data. This reflects real-world LLM scenarios, such as our Healthcare dataset ARMMAN, where reliable reward models and labels are unavailable. While the more advanced multi-agent methods discussed in Appendix A.1 could be applied when external rewards or labels exist, doing so is orthogonal to our contribution. After applying those methods, one can still apply our higher-order information aggregation layer to further improve accuracy, making it **complementary rather than competitive**.
>
> [3] Eray Can Elumar, Cem Tekin, and Osman Yagan. Cost-aware llm-based online dataset annotation.
> arXiv preprint arXiv:2505.15101, 2025.
>
> [4] Vighnesh Subramaniam, Yilun Du, Joshua B Tenenbaum, Antonio Torralba, Shuang Li, and Igor
> Mordatch. Multiagent finetuning: Self improvement with diverse reasoning chains. arXiv preprint
> arXiv:2501.05707, 2025.

---

> ### Author Response · Authors · 2025-11-19
>
> Re W3 **Experimental details**: To address the reviewer’s concern, we now clarify and expand the discussion of our experimental setup. We used vLLM for all inference experiments and adopted its default random seed (0). As already included in the supplemental material, our full codebase and evaluation scripts enable complete reproduction of the experimental environment.
> In addition, we now explicitly report variance estimates across tasks. Specifically, we computed the standard error of overall accuracy for each dataset and aggregation method: approximately 0.19% for UltraFeedback, 0.09% for MMLU, and 0.33% for ARMMAN. These small variances confirm the stability of our results.
> We also added a new set of experiments to further provide a per-question comparison between our methods and the baselines in Table 4. Specifically, for each dataset, we report: (1) the number of questions where majority voting (MV) was incorrect but our method was correct, and vice versa; and (2) the number of questions where the single best (SB) model was incorrect but our method was correct, and vice versa. The total numbers of evaluated questions are $n_{\text{UltraFeedback}} = 56{,}380$, $n_{\text{MMLU}} = 109{,}820$, and $n_{\text{ARMMAN}} = 11{,}785$. We further conduct hypothesis testing for three datasets to compare the performance of our methods (e.g., OW-I) and MV. The resulting t-statistics are 12.53 (UltraFeedback), 23.39 (MMLU), and 3.22 (ARMMAN), indicating that our aggregation methods are **significantly better** than MV.
>
> | Dataset                                                      | Model               | MV wrong → Ours correct / Ours wrong → MV correct | SB wrong → Ours correct / Ours wrong → SB correct |
> | ------------------------------------------------------------ | ------------------- | ------------------------------------------------- | ------------------------------------------------- |
> | **UltraFeedback**                                            | OW-L                | 2545 / 1727                                       | 3281 / 2984                                       |
> |                                                              | OW-I                | 2545 / 1727                                       | 3281 / 2984                                       |
> |                                                              | ISP                 | 2369 / 1762                                       | 3838 / 3773                                       |
> | **MMLU**                                                     | OW-L                | 1821 / 659                                        | 1726 / 2440                                       |
> |                                                              | OW-I                | 1821 / 659                                        | 1726 / 2440                                       |
> |                                                              | ISP                 | 1522 / 667                                        | 1840 / 2956                                       |
> | **ARMMAN**                                                   | OW-L                | 264 / 195                                         | 179 / 124                                         |
> |                                                              | OW-I                | 264 / 195                                         | 179 / 124                                         |
> |                                                              | ISP                 | 264 / 195                                         | 179 / 124                                         |
>
>
> Re Q1 **Correlation between OW-L and OW-I**: OW-L and OW-I are inherently correlated in both parts. On one hand, they both use second-order information to estimate accuracy $x_i$, though in different methods. On the other hand, after estimating $x_i$, both of them take the same equation in Algorithm 1 to compute the optimal weight.
> Moreover, theoretically, as discussed around SP Lines 273–274 and in the finite-sample error bound of Theorem 3, when $N,M\rightarrow \infty$, the estimation error will vanish. Both approaches recover the same optimal weights and therefore yield identical results. Our experiments show that even with relatively small $N$, the two estimators produce very similar weight vectors, leading to nearly identical empirical performance.
> This empirical consistency supports the robustness of our methods and aligns with the theoretical convergence guarantees, confirming that both estimators reliably capture the same underlying aggregation principle.

---

> > ### Comment · Reviewer_96ii · 2025-11-25
> > **Current Remarks**
> >
> > I acknowledge the response to Q1 and am satisfied with the authors' answer. However, I would like to emphasize the remaining concerns regarding the paper's weaknesses and await a more suitable response.

---

> ### Comment · Reviewer_96ii · 2025-11-25
>
> ## **Re (Re W1)**
>
> Thank you for the clarification regarding the main model versus the generalized framework.
>
> I have reviewed the derivation in **Appendix G.1 (Proof of Theorem 4)** and acknowledge that for the Bayesian optimal aggregator, the question difficulty parameter ($\alpha$) effectively cancels out of the optimization objective via the expectation over $D(\alpha)$. Mathematically, I accept that this removes the need to explicitly model specific difficulty levels for each question.
>
> However, this mathematical cancellation does not resolve the fundamental validity concerns regarding the assumptions used to apply this framework to LLMs. While $\alpha$ drops out, the method simply shifts the burden entirely to the **model ability parameter ($\beta_i$)**, and I remain unconvinced by the proposed methods for estimating this parameter.
>
> ### **The estimation methods rely on assumptions the authors already admit are weak**
> To address the lack of ground truth, the paper proposes estimating these parameters (or their equivalent accuracies $x_i$) using second-order information via **OW-L** and **OW-I** in **Section 5**. However, both estimation strategies rely on **Assumption 1 (Conditional Independence)**, which creates a critical validity gap.
>
> * **Critique of OW-L:**
>     Crucially, the failure of Assumption 1 is rooted in how the model defines error. **Proposition 1** explicitly defines the probability of error as uniform across all incorrect options: $\mathbb{P}(A_i = s_k | S^* = s_j) = \frac{1-x_i}{K-1}$. This assumes that when an agent makes a mistake, it picks a wrong answer **purely by random chance**. Recent empirical work explicitly invalidates this for LLMs [1]. Additionally, because the **OW-L** estimation (expanded in **Appendix F.2**) derives its validity from this uniform error assumption, its objective function lacks any term for **correlated error**. Consequently, if two agents fall for the same specific "distractor" (a common LLM failure mode such as bias or toxicity [2, 3]), the model mathematically misinterprets this non-random agreement as high accuracy.
>
> * **Critique of OW-I (ISP-based):**
>     Similarly, the **OW-I** method relies on Inverse Surprising Popularity (ISP) to generate a pseudo-ground truth. However, the theoretical guarantee for ISP (**Theorem 3**) explicitly states that it outperforms majority voting only *"Under Assumption 1."* The advantage function for ISP (**Equation 5**) relies on a "predicted popularity" score calculated by summing probabilities as if agents were independent. If errors are correlated (e.g., due to difficulty spikes), the predicted popularity becomes inaccurate, causing the advantage function and thus the pseudo-ground truth to fail.
>
> The framework presents a mathematically elegant solution for a system where agents are independent and errors are uniform random noise. However, this work is positioned in the LLM setting, where agents share training data and errors are highly correlated and non-uniform [1, 2, 3]. Because the estimation methods in **Section 5** and the error model in **Proposition 1** do not account for these specific properties of LLMs, I am still not convinced that the "optimality" claims hold in practice.
>
> To resolve this, the model used in this work must move beyond the proposition that mistakes are random chance. A robust solution would explicitly model the **correlation structure** between agents rather than assuming a uniform error distribution. This would allow the aggregator to distinguish between "independent consensus" (true signal) and "correlated systematic error."
>
> [1] Elliot Kim, Avi Garg, Kenny Peng, and Nikhil Garg. Correlated errors in large language models. In *International Conference on Machine Learning (ICML)*, 2025.
>
> [2] Kremena Valkanova and Pencho Yordanov. Irrelevant alternatives bias large language model hiring decisions. In *Findings of the Association for Computational Linguistics: EMNLP 2024*, pp. 6899–6912. Association for Computational Linguistics, 2024.
>
> [3] Guiming Hardy Chen, Shunian Chen, Ziche Liu, Feng Jiang, and Benyou Wang. Humans or LLMs as the judge? A study on judgement bias. In *Proceedings of the 2024 Conference on Empirical Methods in Natural Language Processing (EMNLP)*, pp. 13867–13886. Association for Computational Linguistics, 2024.

---

> ### Comment · Reviewer_96ii · 2025-11-25
>
> ## **Re (Re W2)**
>
> I appreciate the authors' clarification regarding the unsupervised nature of the setting. However, I believe that benchmarking against additional standard unsupervised baselines would significantly strengthen the empirical evaluation and contextualize the contribution.
>
> ### **Inclusion of Foundational Unsupervised Aggregation (Dawid-Skene)**
> The response emphasizes the novelty of the "aggregation step" in unsupervised settings. In this context, the standard baseline for accuracy estimation without ground truth is typically **Expectation-Maximization (EM) based methods**, such as **Dawid-Skene [1]**.
> Like the proposed OW-L and OWL-ISP, Dawid-Skene estimates agent expertise purely from **observed agreement patterns**. Since this paper focuses on estimation methods with unknown ground truth, a comparison against the foundational EM algorithm, which solves the same mathematical problem, is a critical baseline to demonstrate advancement. Additionally, I noted that the empirical results for **Surprising Popularity (SP)** on the real-world **ARMMAN** dataset appear to be missing.
>
> ### **Comparison with Unsupervised Multi-Agent Reasoning Strategies**
> The response suggests that advanced multi-agent strategies require "external reward models" or "the labels to exist". However, I would like to point out that many recent strategies, such as **Multi-Agent Debate [2,3]**, operate in a fully unsupervised manner. These methods rely on inter-agent consistency and iterative refinement rather than external labels.
> Comparing against these interaction-based baselines would help justify the benefits of the proposed aggregation approach over simply allowing agents to refine their own consensus, which represents the current state-of-the-art for unsupervised multi-agent reasoning.
>
> [1] A. P. Dawid and A. M. Skene. Maximum likelihood estimation of observer error-rates using the EM algorithm. *Applied Statistics*, 28(1):20–28, 1979.
>
> [2] Tian Liang, Zhiwei He, Wenxiang Jiao, Xing Wang, Yan Wang, Rui Wang, Yujiu Yang, Zhaopeng Tu, and Shuming Shi. Encouraging divergent thinking in large language models through multi-agent debate. *Proceedings of the 2024 Conference on Empirical Methods in Natural Language Processing (EMNLP)*, 2024.
>
> [3] Yilun Du, Shuang Li, Antonio Torralba, Joshua B. Tenenbaum, and Igor Mordatch. Improving factuality and reasoning in language models through multiagent debate. *International Conference on Machine Learning (ICML)*, 2024.

---

> ### Comment · Reviewer_96ii · 2025-11-25
>
> ## **Re (Re W3)**
>
> I appreciate the authors' transparency regarding the single random seed and the inclusion of standard error calculations. This additional data is helpful. However, to fully assess the reproducibility and robustness of the proposed methods, I am specifically looking for a different type of statistical significance than what was provided.
>
> **1. Distinction Between Dataset Confidence and Algorithmic Stability**
> The reported standard errors (e.g., ~0.19%) are quite low, but it is unclear if these values capture the variance of the **method itself** or merely the variance of the data sample.
> Because LLM generation is stochastic, a single run with `seed=0` represents just one potential trajectory. To distinguish whether the performance gain is a consistent algorithmic improvement or specific to a particular generation path, I request the **standard deviation of accuracy across multiple independent runs (e.g., 3-5 random seeds)**. This metric is standard for establishing robustness in probabilistic generative models and would provide much stronger evidence of the method's reliability than a single point estimate.
>
> **2. Significance Testing for Paired Outcomes**
> Regarding the t-statistics provided, I note that comparing binary outcomes on the same dataset is a paired comparison. While the high t-values indicate statistical significance due to the large $N$, this is primarily driven by the dataset size. Reporting the variance across seeds (as requested above) would allow for a more rigorous significance test (e.g., a paired t-test on the seed-level means) that isolates the variance of the *algorithm* from the variance of the *data*.
>
> I believe adding these specific stability metrics, standard deviation across seeds. would greatly strengthen the paper's claims of robustness.

---

> ### Author Response · Authors · 2025-11-27
>
> # RE (RE RE W1):
>
> We believe this follow-up comment of W1 comes from several **misunderstandings** of our model. We offer the following point-by-point clarifications and hope they fully resolve your concerns.
>
> Firstly, regarding the assumption of $\alpha,\beta$ for Theorem 4, we want to further emphasize it is totally **unrelated** to our main framework and main theory results(Theorem 1,2,3). As we mentioned in above response to W1, "we do **not** make any assumptions regarding question difficulties or model abilities. In the appendix, we generalized our framework to capture the situation where additional information about question difficulties and model abilities is available." **This should be treated as a possible extension, instead of limitations of our main framework.** Also, we don't really understand what "I remain unconvinced by the proposed methods for estimating this parameter" means because all our estimation methods(OW-L, OW-I) are for the main framework without $\alpha,\beta$ instead of this extension setting.
>
> Secondly, regarding the bold sentence "The estimation methods rely on assumptions the authors already admit are weak", we wish to clarify that we have **never** admitted or stated that our assumptions are unrealistic or problematic. **We kindly ask the reviewer to specify which part of the text led to this conclusion.**
> We also suspect this comment may stem from a misunderstanding of the terminology. In mathematical and theoretical analysis, describing an assumption as "weak" is a positive attribute. It means the condition is mild, unrestrictive, and general. A "weak assumption" is actually a strength of the paper, as it implies the method is applicable to a broader range of scenarios without requiring strict ("strong") constraints. We maintain that our assumptions are natural and easily satisfied in standard settings.
>
> Thirdly, regarding your critique of OW-L and OW-I, the uniform error property is **not** an assumption but a structural result guaranteed by our construction. As detailed in Lines 141-156, we preprocess the answers using random shuffling. This randomization mechanism naturally enforces statistical uniformity on the error distribution over the shuffled indices (Proposition 1). **Crucially, this does not imply that we need to naively treat all incorrect answers as semantically equivalent or equal in the original distribution**; rather, the shuffling merely symmetrizes the noise structure to facilitate estimation. We further prove in Appendix B.1 that this random shuffling preprocessing does not result in information loss within our setting.
>
> Fourthly, we believe the concern regarding Assumption 1 arises from **conflating marginal dependence with conditional independence.** We agree that model outputs are marginally correlated across the dataset (e.g., most models answer easy questions correctly). However, Assumption 1 requires conditional independence given one fixed question. Once conditioned on the specific input and ground truth, the remaining variability in the LLMs' outputs arises from their **internal stochastic generation processes** (e.g., sampling temperature). These stochastic processes are independent across different models. Therefore, while models are correlated over the population of questions, they still satisfy conditional independence.

---

> ### Author Response · Authors · 2025-11-27
>
> # RE (RE RE W2):
>
> Firstly, we appreciate the suggestion to include the Dawid-Skene (DS) algorithm as an additional baseline. We evaluated DS using the identical setup as our main experiments in Table 3. The updated results (see table below) demonstrate that while Dawid-Skene is indeed a stronger baseline than Majority Voting, our methods (OW-L and OW-I) still achieve consistent and noticeable improvements over it.
>
> Secondly, We initially omitted SP experiments because we provide both theoretical analysis (Theorem 2) that the traditional second-order information method SP is inferior to MV and our second-order information–based method ISP, and corresponding explanations (Lines 285-296). Therefore, we viewed it as a less critical algorithm given our goal of outperforming MV. However, for completeness, we have now evaluated the SP algorithm across all three datasets, using the same setup as Table 3. As shown in the updated Table 3, the empirical performance of SP aligns with our theory.
> | Accuracy | OW-L (ours) | OW-I (ours) | ISP (ours) | SP | MV | DS | Single Best |
> | :--- | :---: | :---: | :---: | :---: | :---: | :---: | :---:|
> | **UltraFeedback** | **73.66%** | **73.66%** | 73.26% | 72.25% | 72.21% |73.08%| 73.14% |
> | **MMLU** | **90.37%** | **90.37%** | 90.01% | 88.26% | 89.32% |90.10%| 91.02% |
> | **ARMMAN** | **85.78%** | **85.78%** | **85.78%** | 84.60% | 85.24% | 85.49%| 85.32% |
>
> Thirdly, regarding comparing with interaction-based strategies like Multi-Agent Debate, we wish to clarify that our work operates in a distinct setting. Debate methods rely on iterative, multi-round communication between agents to refine answers, which significantly increases computational cost and latency. Our framework focuses on **one-round, training-free** inference without requiring agent interaction. Also, in a debate framework, the final consensus is often reached via Majority Voting [1]. Our aggregation method could potentially replace MV within a debate pipeline (aggregating the final outputs of debating agents) to further boost performance. **Consequently, our method is orthogonal to debate strategies, not a direct competitor.**
>
> [1] Vighnesh Subramaniam, Yilun Du, Joshua B Tenenbaum, Antonio Torralba, Shuang Li, and Igor Mordatch. Multiagent finetuning: Self improvement with diverse reasoning chains. arXiv preprint arXiv:2501.05707, 2025.
>
>
>
> # RE (RE RE W3):
>
> First, regarding the statistical metrics, the low standard errors and high t-values reflect our method's **consistent superiority** over the baseline across the entire large-scale dataset. This confirms that our performance gains arise from statistical significance in large samples, rather than the randomness or noise often observed in small-sample scenarios.
>
> Also, we wish to clarify that in recent literature involving large-scale LLM inference, it is standard practice to report results based on a single generation seed. This is primarily due to the high computational/API costs associated with re-running inference on massive datasets (e.g., MMLU, UltraFeedback) multiple times.
>
> However, we agree with your insight that evaluating algorithmic robustness is crucial. To address this, we conducted an additional full-scale evaluation using a different random seed (42). As observed, while the absolute accuracy values naturally fluctuate due to LLM stochasticity, **the relative performance pattern remains identical**. Our methods (OW-L/OW-I) **consistently outperform** the baselines (including the newly added SP and DS) in this independent run. This consistency in performance ranking confirms that our improvements are robust and not an artifact of a specific seed.
>
> | Accuracy | OW-L (ours) | OW-I (ours) | ISP (ours) | SP | MV | DS | Single Best |
> | :--- | :---: | :---: | :---: | :---: | :---: | :---: | :---: |
> | **UltraFeedback** | **73.66%** | **73.66%** | 73.02% | 72.00% | 72.09% | 72.57% | 73.29% |
> | **MMLU** | **90.20%** | **90.20%** | 89.90% | 88.33% | 89.26% | 89.98% | 90.78% |
> | **ARMMAN** | **85.96%** | **85.96%** | **85.96%** | 84.99% | 85.41% | 85.70% | 85.78% |

---

> > ### Author Response · Authors · 2025-11-27
> >
> > Thank you again for your insightful review. We believe we have addressed all your concerns and hope you will consider raising the score accordingly.

---

### Meta-Review · Area_Chair_RQy1 · 2025-12-27

**Summary:**

Thanks for the reviewers’ valuable efforts in reviewing this paper and authors’ submission. This paper targets the limitation of existing majority voting in aggregating the responses of large language models (LLMs), and proposes two aggregation algorithms: Optimal Weight and Inverse Surprising Popularity. Experiments conducted on synthetic datasets validate the effectiveness of the proposed methods. Four reviewers’ comments and concerns are summarized as follows:
- Strengths: Reviewer 96ii, 2A2J and 1dqM acknowledge that the paper studies an important and novel problem of how to effectively aggregate the LLM responses. Reviewer TNoU notes that the theoretical analysis is mathematically rigorous. Reviewers 96ii, TNoU, and 1dqM recognize the performance improvements demonstrated in the experimental results. In addition, Reviewer 96ii highlights the potential applicability of the proposed algorithms in unsupervised and real-world scenarios, such as automated data annotation. However, the reviewers also raise concerns as follows:
- Weakness: Reviewers 96ii and 1dqM point out the limited scope of the evaluation benchmarks, not including commonly used LLM reasoning benchmark such as GSM8K. Reviewer 1dqM further raises concerns regarding the insufficient and incomplete discussion of related work. Reviewer TNoU questions the trade-off between the increased complexity of the proposed methods and the relatively limited empirical improvements. Reviewer 96ii raises the concerns about the experimental details and reproducibility. Reviewers TNoU and 2A2J also express concerns about the generalizability of the proposed methods, particularly their extension to open-ended tasks or unseen datasets. Reviewer 2A2J points out the wrong description of the MoE in this paper.

Ratings: The ratings of the four reviewers are 4, 4, 4, and 4, respectively. All ratings are below the acceptance threshold. Considering the negative consensus among the reviewers, I recommend a reject.

**Reviewer Concerns:**

During the rebuttal period, the authors partially addressed the reviewers’ concerns. For instance, reviewer 96ii expressed satisfaction with the authors’ clarification regarding the question about the correlation between OW-L and OW-I. Reviewer 1dqM also increased the presentation score to 3 following the rebuttal.

However, most major concerns remain insufficiently unsolved. For example, reviewer 1dqM explicitly agreed with the reviewer 96ii’s comments and noted that these concerns still require adequate attention. Reviewer 96ii also emphasized the remaining concerns after the authors’ first-round comments. Regarding concerns about evaluation scope and generalization, the authors did not provide additional benchmarks or experimental results during the rebuttal. Overall, the paper still leaves substantial room for further improvement.

Additionally, all four reviewers provided detailed and constructive comments covering multiple aspects, including evaluation, algorithm complexity, experiments and related work. Notably, reviewer 96ii’s key concerns were echoed and supported by reviewer 1dqM.

**Reviewer Scores:**

The ratings of the four reviewers are 4, 4, 4, and 4, respectively. Reviewer 96ii raised the presentation score to 3 during the rebuttal. In addition to this, there were no rating changes during the rebuttal period. Although reviewers 96ii and 1dqM indicated a potential willingness to change their scores and were waiting for further discussion from the authors, the authors did not fully address the major concerns. The other two reviewer TNoU and 2A2J did not respond to the authors’ comments. Overall, all four reviewers reached a negative consensus with rating of 4 for this submission.

---

### Decision · Program_Chairs · 2026-01-26

Reject